# RETRIEVAL-AUGMENTED
# REINFORCEMENT LEARNING

## ABSTRACT

Most deep reinforcement learning (RL) algorithms distill experience into parametric behavior policies or value functions via gradient updates. While effective, this approach has several disadvantages: (1) it is computationally expensive, (2) it can take many updates to integrate experiences into the parametric model, (3) experiences that are not fully integrated do not appropriately influence the agent's behavior, and (4) behavior is limited by the capacity of the model. In this paper we explore an alternative paradigm in which we train a network to map a dataset of past experiences to optimal behavior. Specifically, we augment an RL agent with a retrieval process (parameterized as a neural network) that has direct access to a dataset of experiences. This dataset can come from the agent's past experiences, expert demonstrations, or any other relevant source. The retrieval process is trained to retrieve information from the dataset that may be useful in the current context, to help the agent achieve its goal faster and more efficiently. We integrate our method into two different RL agents: an offline DQN agent and an online R2D2 agent. In offline multi-task problems, we show that the retrieval-augmented DQN agent avoids task interference and learns faster than the baseline DQN agent. On Atari, we show that retrieval-augmented R2D2 learns significantly faster than the baseline R2D2 agent and achieves higher scores. We run extensive ablations to measure the contributions of the components of our proposed method.

## 1 INTRODUCTION

A host is preparing a holiday meal for friends. They remember that the last time they went to the grocery store during the holiday season, all of the fresh produce was sold out. Thinking back to this past experience, they decide to go early! The hypothetical host is employing *case-based reasoning* (e.g., Kolodner, 1992; Leake, 1996). Here, an agent *recalls* a situation similar to the current one and uses information from the previous experience to solve the current task. This may involve adapting old solutions to meet new demands, or using previous experiences to make sense of new situations.

In contrast, a dominant paradigm in modern reinforcement learning (RL) is to learn general purpose behaviour rules from the agent's past experience. These rules are typically represented in the weights of a parametric policy or value function network model. Most deep RL algorithms integrate information *across trajectories* by iteratively updating network parameters using gradients that are computed along *individual trajectories* (collected online or stored in an experience replay dataset, Lin, 1992). For example, many off-policy algorithms reuse past experience by "replaying" trajectory snippets in order to compute weight updates for a value function represented by a deep network (Ernst et al., 2005; Riedmiller, 2005; Mnih et al., 2015b; Heess et al., 2015; Lillicrap et al., 2015).

This paradigm has clear advantages but at least two interrelated limitations: First, after learning, an agent's past experiences no longer plays a direct role in the agent's behavior, even if it is relevant to the current situation. This occurs because detailed information in the agent's past experience is lost due to practical constraints on network capacity. Second, since the information provided by individual trajectories first needs to be distilled into a general purpose parametric rule, an agent may not be able to exploit the specific guidance that a handful of individual past experiences could provide, nor rapidly incorporate novel experience that becomes available—it may take many replays through related traces in the past experiences for this to occur (Weisz et al., 2021).

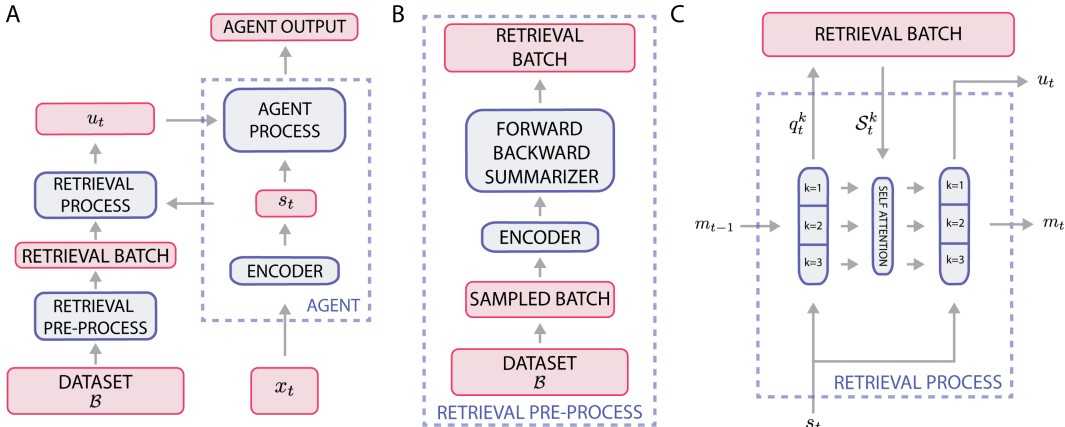

Figure 1: **Retrieval-augmented agent (R2A) architecture**: (A) R2A augments the agent with a retrieval process. The retrieval process and the agent maintain separate internal states, $\boldsymbol{m}_t$ and $\boldsymbol{s}_t$, respectively. The retrieval process retrieves information relevant to the agent's current internal state $\boldsymbol{s}_t$ from the retrieval batch, which is a pre-processed sample from the retrieval dataset $\mathcal{B}$. The retrieved information $\boldsymbol{u}_t$ is used by the agent process to inform its output (e.g., a policy or value function). (B) A batch of raw trajectories is sampled from the retrieval dataset $\mathcal{B}$ and encoded (using the same encoder as the agent). Each encoded trajectory is then summarized via forward and a backward summarization functions (section 2.2) and sent to the retrieval process. (C) The retrieval process is parameterized as a recurrent model and the internal state $\boldsymbol{m}_t$ is partitioned into slots. Each slot independently retrieves information from the retrieval batch, which is used to update the slot's representation and sent to the agent process in $\boldsymbol{u}_t$. Slots also interact with each other via self-attention. See section 2.3 for more details.

In this work, we develop an algorithm that overcomes these limitations by *augmenting* a standard reinforcement learning agent with a *retrieval process* (parameterized via a neural network). The purpose of the retrieval process is to help the agent achieve its objective by providing relevant contextual information. To this end, the retrieval process uses a learned attention mechanism to dynamically access a large pool of past trajectories stored in a dataset (e.g., a replay buffer), with the aim of integrating information across these. The proposed algorithm (R2A), shown in Figure 1, enables an agent to retrieve information from a dataset of trajectories. The high-level idea is to have two different processes. First, the *retrieval process*, makes a "query" to search for relevant contextual information in the dataset. Second, the *agent process* performs inference and learning based on the information provided by the retrieval process. These two processes have different internal states but interact to shape the representations and predictions of each other: the agent process provides the relevant context, and the retrieval process uses the context and its own internal state to generate a query and retrieve relevant information, which is in turn used by the agent process to shape the representation of its policy and value function (see Fig. 1A). Our proposed retrieval-augmented RL paradigm could take several forms. Here, we focus on a particular instantiation to assay and validate our hypothesis that learning a retrieval process can help an RL agent achieve its objectives.

**Summary of experimental results.** We want RL algorithms that are able to adapt to the available data source and usefully ingest any dataset. Hence, we test the performance of the proposed method in three different scenarios. First, we evaluate it on Atari games in a single task setting. We build upon R2D2 (Kapturowski et al., 2018), a state-of-the-art off-policy RL algorithm. Second, we evaluate it on a multi-task offline RL environment, using DQN (Mnih et al., 2013) as the RL algorithm, where the data in the queried dataset belongs to the same task. Third, we evaluate it on a multi-task offline RL environment where the data in the dataset comes from multiple tasks. In all these cases, we show that R2A learns faster and achieves higher reward compared to the baseline.

## 2 RETRIEVAL-AUGMENTED AGENTS

We now present our method for augmenting an RL agent with a retrieval process, thereby reducing the agent's dependence on its model capacity, and enabling fast and flexible use of past experiences. A retrieval-augmented agent (R2A) consists of two main components: (1) the retrieval process, which takes in the current state of the agent, combines this with its own internal state, and retrieves relevant information from an external dataset of experiences; and (2) a standard reward-maximizing RL agent, which uses the retrieved information to improve its value or policy estimates. See Figure 1 for an overview. The retrieval process is trained to retrieve information that the agent can use to improve its performance, without explicit knowledge of the agent's policy. Importantly, the retrieval process has its own internal state, which enables it to integrate and combine information across

retrievals. In the following, we focus on value-based methods, such as DQN (Mnih et al., 2015a) and R2D2 (Kapturowski et al., 2018), but our approach is equally applicable to policy-based methods.

## 2.1 RETRIEVAL-AUGMENTED AGENT

Formally, the agent receives an input $x_t$ at each timestep $t$. Each input is processed by a neural encoder (e.g., a resnet if the input is an image) to obtain an abstract internal state for the agent $s_t = f_\theta^{\text{enc}}(x_t)$. For clarity, we focus here on the case of a single vector input, however, each input could also include the history of past observations, actions, and rewards, as is the case when $f_\theta^{\text{enc}}$ is a recurrent network. These embeddings are used by the agent and retrieval processes. The retrieval process operates on a dataset $\mathcal{B} = \{(x_t, a_t, r_t), \dots, (x_{t+T}, a_{t+T}, r_{t+T})\}$ of $l$-step trajectories, for $l \geq 1$. This dataset could come from other agents or experts, as in offline RL or imitation learning, or consist of the growing set of the agent's own experiences. Then, a retrieval-augmented agent (R2A) consists of the retrieval process and the agent process, parameterized by $\theta = \{\theta^{\text{enc}}, \theta^{\text{retr}}, \theta^{\text{agent}}\}$,

$$\textbf{Retrieval process } f_{\theta,\mathcal{B}}^{\text{retr}} : m_{t-1}, s_t \mapsto m_t, u_t$$

$$\textbf{Agent process } f_\theta^{\text{agent}} : s_t, u_t \mapsto Q_\theta(s_t, u_t, a)$$

***Retrieval Process.*** The retrieval process is parameterized as a neural network and has an internal state $m_t$. The retrieval process takes in the current abstract state of the agent process $s_t$ and its own previous internal state $m_{t-1}$ and uses these to retrieve relevant information from the dataset $\mathcal{B}$, which it then summarizes in a vector $u_t$, and also updates its internal state $m_t$.

***Agent Process.*** The current state of the agent $s_t$ and the information from the retrieval process $u_t$ is then passed to the action-value function, itself used to select external actions.

The above defines a parameterization for a retrieval-augmented agent. For retrieval to be effective, the retrieval process needs to: (1) be able to efficiently query a large dataset of trajectories, (2) learn and employ a similarity function to find relevant trajectories, and (3) encode and summarize the trajectories in a manner that allows efficient discovery of relevant past and future information.

Below, we explain how we achieve these desiderata. At a high-level, to reduce computational complexity given a experience dataset of hundreds of thousands of trajectories, R2A operates on samples from the dataset. R2A then encodes and summarizes the trajectories in these samples using auxiliary losses and bi-directional sequence models to enable efficient retrieval of temporal information. Finally, R2A uses attention to select semantically relevant trajectories.

## 2.2 RETRIEVAL BATCH SAMPLING AND PRE-PROCESSING.

***Sampling a retrieval batch from the retrieval dataset.*** To reduce the computational complexity, R2A uniformly samples a large batch of past experiences from the retrieval dataset, and only uses the sampled batch for retrieving information. We denote the sampled batch as "retrieval batch" and the number of trajectories in the retrieval batch as $n_{\text{retrieval}}$.

***Encoding and forward-backward summarization of the retrieval dataset and corresponding auxiliary losses.*** Since the agent's internal state extracts information from observations which relate to the task at hand, we choose to re-encode the raw experiences in the "retrieval batch" using the agent encoder module (i.e., $f_\theta^{\text{enc}}$). However, this representation is a function only of past observations (i.e., it's a causal representation) and may not be fully compatible with the needs of the retrieval operation. For that reason, we propose to further encode the retrieved batch of information, by additionally learning a *summarization* function, applied on the output of the encoder module, and which captures information about the past and the future within a particular trajectory by using a bi-directional model (e.g., parameterized as a bi-directional RNN or a Transformer).

$$\textbf{Forward Summarizer } f_\theta^{\text{fwd}} : (s_1, \dots, s_t) \mapsto h_t$$

$$\textbf{Backward Summarizer } f_\theta^{\text{bwd}} : (s_T, \dots, s_t) \mapsto b_t$$

For each trajectory in the retrieval batch, we represent each time-step within a trajectory by a set of two vectors $h_{i,t}$ and $b_{i,t}$ (Figure 6 in the appendix) where $h_{i,t}$ summarizes the past (i.e., from

---

**Algorithm 1** One timestep of a retrieval-augmented agent (R2A).

---

***Input:*** Current input $\boldsymbol{x}_t$, previous retrieval process state $\boldsymbol{m}_{t-1} = \{\boldsymbol{m}_{t-1,k} | \, k \in \{1, \ldots, n_f\}\}$,
dataset of $l$-step trajectories $\mathcal{B} = \{(\boldsymbol{x}_t^i, \boldsymbol{h}_t^i, \boldsymbol{b}_t^i, a_t^i, r_t^i) \ldots (\boldsymbol{x}_{t+l}^i, \boldsymbol{h}_{t+l}^i, \boldsymbol{b}_{t+l}^i, a_{t+l}^i, r_{t+l}^i)\}$
for $l \geq 1$ and $1 \leq i \leq n_{\text{traj}}$, where $\boldsymbol{h}$ and $\boldsymbol{b}$ are the outputs of the forward & backward summarizers.

**Encode the current input at time-step $t$.**
$\boldsymbol{s}_t = f_\theta^{\text{enc}}(\boldsymbol{x}_t)$

**Step 1: Compute the query.** For all $1 \leq k \leq n_f$, compute
$\widehat{\boldsymbol{m}}_{t-1}^k = \text{GRU}_{\boldsymbol{\theta}}\left(\boldsymbol{s}_t, \boldsymbol{m}_{t-1}^k\right)$
$\boldsymbol{q}_t^k = f_{\text{query}}(\widehat{\boldsymbol{m}}_{t-1}^k)$

**Step 2: Identify the most relevant trajectories.** For all $1 \leq k \leq n_f, 1 \leq j \leq l$ and $1 \leq i \leq n_{\text{traj}}$,
$\boldsymbol{\kappa}_{i,j} = (\boldsymbol{h}_j^i \boldsymbol{W}_{\text{ret}}^{\text{e}})^{\text{T}}$
$\ell_{i,j}^k = \left(\frac{\boldsymbol{q}_t^k \boldsymbol{\kappa}_{i,j}}{\sqrt{d_e}}\right)$
$\alpha_{i,j}^k = \text{softmax}\left(\ell_{i,j}^k\right)$.
Given scores $\alpha$, the top-$k_{\text{traj}}$ trajectories (resp. top-$k_{\text{states}}$ states) are selected and denoted by $\mathcal{T}_t^k$ (resp. $\mathcal{S}_t^k$).

**Step 3: Retrieve information from the most relevant trajectories and states.**
$\alpha_{i,j}^k = \text{softmax}\left(\ell_{i,j}^k\right), i \in \mathcal{T}_t^k, j \in \mathcal{S}_t^k$.
$\boldsymbol{g}_t^k = \sum_{i,j} \alpha_{i,j}^k \boldsymbol{v}_{i,j}$ where $\boldsymbol{v}_{i,j} = \boldsymbol{b}_{i,j} \boldsymbol{W}_{\text{ret}}^{\text{v}}$

**Step 4: Regularize the retrieved information by using information bottleneck.**
$\boldsymbol{z}_t^k \sim p(z | \boldsymbol{g}_t^k)$

**Step 5: Update the states of the slots.**
Slotwise update using retrieved information:
$\widetilde{\boldsymbol{m}}_t^k \leftarrow \widehat{\boldsymbol{m}}_{t-1}^k + \boldsymbol{z}_t^k \quad \forall k \in \{1, \ldots, n_f\}$
Joint slot update through self-attention:
$\boldsymbol{c}_t^k = \widehat{\boldsymbol{m}}_{t-1}^k \boldsymbol{W}_{\text{SA}}^q \quad \forall k\{1, \ldots, n_f\}$
$\beta_{k,k'} = \text{softmax}_{k'}\left(\frac{\boldsymbol{c}_t^k \boldsymbol{\kappa}_t^{k'}}{\sqrt{d_e}}\right)$ where $\boldsymbol{\kappa}_t^{k'} = (\widetilde{\boldsymbol{m}}_t^{k'} \boldsymbol{W}_{\text{SA}}^e)^{\text{T}} \quad \forall k, \, k' \in \{1, \ldots, n_f\}$
$\boldsymbol{m}_t^k \leftarrow \widetilde{\boldsymbol{m}}_t^k + \sum_{k'} \beta_{k,k'} \boldsymbol{v}_{k'}$ where $\boldsymbol{v}_{k'} = \widetilde{\boldsymbol{m}}_t^k \boldsymbol{W}_{\text{SA}}^v \quad \forall k \in \{1, \ldots, n_f\}$

**Step 6: Update the agent state using the retrieved information.**
$\boldsymbol{d}_t = \boldsymbol{s}_t \boldsymbol{W}_{\text{ag}}^q$
$\boldsymbol{\kappa}^k = (\boldsymbol{z}_t^k \boldsymbol{W}_{\text{ag}}^e)^{\text{T}} \quad \forall k \in \{1, \ldots, n_f\}$
$\gamma_k = \text{softmax}_k\left(\frac{\boldsymbol{d}_t \boldsymbol{\kappa}^k}{\sqrt{d_e}}\right)$
$\boldsymbol{u}_t \leftarrow \sum_k \gamma_k \boldsymbol{v}_k$ where $\boldsymbol{v}_k = \boldsymbol{z}_t^k \boldsymbol{W}_{\text{ag}}^v \quad \forall k \in \{1, \ldots, n_f\}$.
$\widetilde{\boldsymbol{s}}_t \leftarrow \boldsymbol{s}_t + \boldsymbol{u}_t$

---

$t' = 0$ to $t' = t$ time-steps of the $i^{\text{th}}$ trajectory) while $\boldsymbol{b}_{i,t}$ summarizes the future (i.e., from $t' = t$ to $t' = \ell$ time-steps) within the $i^{\text{th}}$ trajectory. In addition, taking inspiration from (Jaderberg et al., 2016; Trinh et al., 2018; Ke et al., 2019; Devlin et al., 2018; Mazoure et al., 2020; Banino et al., 2021), we use auxiliary losses to improve modeling of long term dependencies when training the parameters of our forward and backward summarizers. The goal of these losses is to force the representation $(\boldsymbol{h}_{i,t}, \boldsymbol{b}_{i,t})_{i,t \geq 0}$ to capture meaningful information for the unknown downstream task. For our experiments, we use supervised losses where we have access to actions or rewards in the retrieval batch. For ablations we also experiment with self-supervised losses. For supervised auxiliary losses, we use policy, value and reward prediction (Silver et al., 2017; Schrittwieser et al., 2019), and for self-supervised losses, we use a BERT-style masking loss (Devlin et al., 2018).

## 2.3 RETRIEVING CONTEXTUAL INFORMATION FROM PAST EXPERIENCES.

In this section, we explain how the retrieval process, when provided with relevant contextual information represented by the agent's current state $\boldsymbol{s}_t$, interacts with the summarized information in the retrieval batch to select information $\boldsymbol{u}_t$ to provide to the agent in return.

**Retrieval process state parameterization.** We parameterize the process that retrieves information from past experience as a structured parametric model with multiple separate memory slots (or sub-units). The state of the retrieval process is a set of $n_f$ memory slots denoted by $\boldsymbol{m}_t = \{\boldsymbol{m}_t^k | \ k \in \{1, \ldots, n_f\}\}$ (indexed by the agent time-step $t$). Slots are initialized randomly at the beginning of the episode. Each slot independently queries and retrieves relevant information from the pool of data. Subsequently the slots update their values independently based on the retrieved information. This is followed by an integration step during which information is shared between slots. Algorithm 1 provides a specification of the six steps of R2A which we now explain in detail below.

**Step 1: Query computation.** Each slot independently computes its prestate using an GRU on the contextual information from the agent: $\widehat{\boldsymbol{m}}_{t-1}^k = \mathrm{GRU}_{\boldsymbol{\theta}}\left(\boldsymbol{s}_t, \boldsymbol{m}_{t-1}^k\right) \quad \forall k \in \{1, \ldots, n_f\}$. Then, each slot independently computes a *retrieval query* which will be matched against information in the retrieval batch: $\boldsymbol{q}_t^k = f_{\mathrm{query}}(\widehat{\boldsymbol{m}}_{t-1}^k) | \ k \in \{1, \ldots, n_f\}\}$[1] where $\boldsymbol{q}_t^k$ is the query generated by the $k^{\mathrm{th}}$ slot at timestep $t$.

**Step 2: Identification of most relevant trajectories and states for each slot (Figure 5A).** The retrieval mechanism process uses an attention mechanism to match a query produced by the retrieval state associated with each slot $m_t^k$ to keys computed on each time step of each trajectory of the retrieval batch. Formally, for each time step and each trajectory in the buffer, we compute a key $\boldsymbol{\kappa}_{i,j}$ by using a linear projection with matrix $\boldsymbol{W}_{\mathrm{ret}}^{\mathrm{e}}$ on the forward summaries $h$: $\boldsymbol{\kappa}_{i,j} = (\boldsymbol{h}_j^i \boldsymbol{W}_{\mathrm{ret}}^{\mathrm{e}})^{\mathrm{T}}$. Each query $\boldsymbol{q}_t^k$ is then matched with the set of all keys $\boldsymbol{\kappa}_{i,j}$, forming attention logits[2] $\ell_{i,j}^k = \left(\frac{\boldsymbol{q}_t^k \boldsymbol{\kappa}_{i,j}}{\sqrt{d_e}}\right)$ and corresponding attention weights $\alpha_{i,j}^k = \mathrm{softmax}\left(\ell_{i,j}^k\right), i \leq n_{\mathrm{traj}}, 0 \leq j \leq T$.

Intuitively, $\alpha_{i,j}^k$ captures the extent to which the $j^{\mathrm{th}}$ timestep of the $i^{\mathrm{th}}$ trajectory in the buffer will be relevant to memory $m_t^k$ through the query $q_t^k$. It follows that $\sum_j \alpha_{i,j}^k$ is a measure of how relevant the $i^{\mathrm{th}}$ trajectory is as a whole for $q_t^k$. Following previous work (Ke et al., 2018; Goyal et al., 2019b), matching only on the most relevant trajectories will increase the robustness of the retrieval mechanism. We therefore select, for each query, the set $\mathcal{T}_t^k$ of $k_{\mathrm{traj}}$ trajectories with highest aggregated score $\sum_j \alpha_{i,j}^k$. Note that typically the queries corresponding to different slots will select different top-$k_{\mathrm{traj}}$ trajectories from the retrieval batch. Following the selection of relevant trajectories, we renormalize the weights $\alpha$, and use a top-$k$ mechanism once-again, this time to choose the set of most relevant states $\mathcal{S}_t^k$ (i.e. which maximizes $\sum_{i \in \mathcal{T}_t^k} \alpha_{i,j}$).

**Step 3: Information retrieval from the most relevant trajectories and states (Figure 5B).** The next step of the retrieval mechanism consists in computing the renormalized weights $\alpha$ on the subsets $\mathcal{T}_t^k$ and $\mathcal{S}_t^k$ ($\alpha_{i,j}^k = \mathrm{softmax}\left(\ell_{i,j}^k\right), i \in \mathcal{T}_t^k, j \in \mathcal{S}_t^k$) and use those weights to compute the final retrieved information. The value retrieved from the buffer for query $q_t^k$ is computed as the $\alpha$-weighted average of a linear function of the *backward* state summaries: $\boldsymbol{g}_t^k = \sum_{i,j} \alpha_{i,j}^k \boldsymbol{v}_{i,j}$ where $\boldsymbol{v}_{i,j} = \boldsymbol{b}_{i,j} \boldsymbol{W}_{\mathrm{ret}}^{\mathrm{v}}$.

**Step 4: Regularization of the retrieved information via an information bottleneck.** We regularize the retrieved information $\boldsymbol{g}_t^k$ via the use of an information bottleneck (Tishby et al., 2000; Alemi et al., 2016); intuitively, each query pays a price of to exploit information from the retrieval batch. Formally, we parametrize two Gaussian distributions $p(Z|g_t^k)$ (which has access to the retrieved information) and $r(Z|m_{t-1})$ (which only has access to the memory units). We define $\boldsymbol{z}_t^k$ as a single sample from $p(Z|g_t^k)$ via the reparameterization trick to ensure differentiability (Kingma & Welling, 2013; Rezende et al., 2014), and ensure that $z_t^k$ does not contain too much information by adding an additional loss $D_{\mathrm{KL}}(p||r)$ to the overall agent loss. We provide more details in the appendix.

**Step 5: Slot update.** The representation of each slot is first additively updated as a function of the retrieved information $\widehat{\boldsymbol{m}}_t^k \leftarrow \widehat{\boldsymbol{m}}_{t-1}^k + \boldsymbol{z}_t^k$. The final update $m_t^k$ consists in an update in which all slots interact through self-attention (as normally done in Transformers) (see Algorithm 1 for details).

**Step 6: Updating the agent state using retrieved information.** The primary goal of the retrieval process is to extract information which may be useful for the agent process. Here, we use the retrieved information to change the state of the agent $s_t$. In the previous step, the retrieved information is used to change the state of the slots. For this step, we use a similar attention mechanism. Here, the

---

[1] $f_{\mathrm{query}}$ is parameterized as a neural network.
[2] We drop time indexing from attention-related quantities to simplify notation.

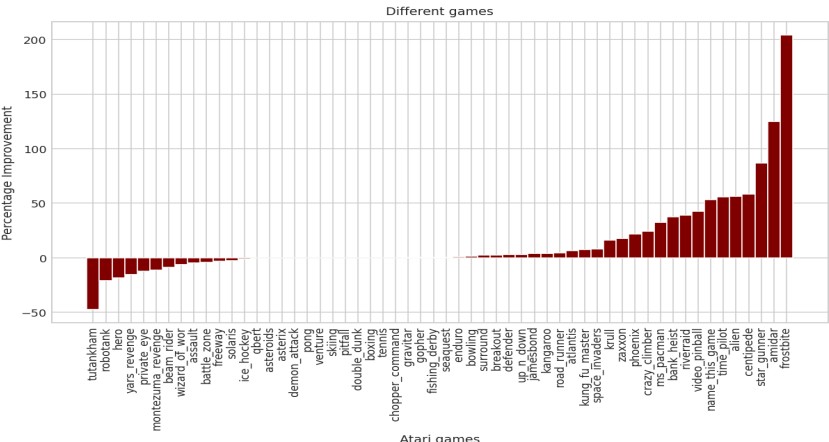

Figure 2: Relative improvement of retrieval-augmented R2D2 vs vanilla R2D2 on different Atari games, measured by human normalized score. We report the average score from 3 seeds per method and per game.

query is a function of the state of the agent process $d_t = s_t W_{\text{ag}}^q$, which are matched with the keys $\kappa^k = (z_t^k W_{\text{ag}}^e)^{\text{T}} \quad \forall k \in \{1, \ldots, n_f\}$, as a result of retrieving information, forming attention weights $\gamma_k = \text{softmax}\left(\frac{d_t \kappa^k}{\sqrt{d_e}}\right)$. The values generated as a result of retrieved information by different slots and the attention weights are then used to update the state of the learning agent : $u_t \leftarrow \sum_k \gamma_k v_k$ where $v_k = z_t^k W_{\text{ag}}^v \quad \forall k \in \{1, \ldots, n_f\}$. $u_t$ is the result of the attention over the retrieved information which is then used to change the representation of the agent process : $\widetilde{s}_t \leftarrow s_t + u_t$. We also shape the representation of the action-value function $Q(s_t, z_t^k, a_t)$ by conditioning the value function on the retrieved information $z_t^k$ (again via a similar attention mechanism).

## 3 EXPERIMENTAL RESULTS

To evaluate R2A, we analyze its performance in three different settings. First, we evaluate on the Atari arcade learning environment (ALE) (Bellemare et al., 2013), a single-task off-policy setting where the retrieval process extracts relevant information from the agent's current replay buffer. We then run a series of ablations of R2A to better understand the roles and effects of its components. Second, we evaluate on a multi-task, offline environment that we created, called *gridroboman*. In this environment, a single network is trained on data from all tasks and then, at evaluation time, the retrieval process queries a retrieval dataset containing only data from the task being evaluated. Third, we evaluate R2A in a multi-task offline version of the BabyAI environment (Chevalier-Boisvert et al., 2018). Again, a single network is trained on all tasks but now the retrieval process queries a retrieval dataset containing data from all tasks.

In our experiments, the retrieval process selects the top $k_{\text{traj}} = 10$ most relevant trajectories (step 2, section 2.3), and then retrieves relevant information from the selected trajectories (step 3, section 2.3) using the top $k_{\text{states}} = 10$ most relevant states. To summarize the experiences in the retrieval batch we use a forward and backward GRU with 512 hidden units. To train the representation of these, we use auxiliary losses in the form of action, reward, and value prediction (section 2.2). See the section A.3 in appendix for more details about the experimental setup and training losses.

### 3.1 ATARI: SINGLE-TASK OFF-POLICY RL

In this first experiment, our goal is to evaluate whether retrieval augmentation improves the performance and sample efficiency of a strong, recurrent baseline agent on a challenging, visually-complex environment—the Atari 2600 videogame suite (Bellemare et al., 2013). We use recurrent replay distributed DQN (R2D2, Kapturowski et al. (2018)) as the baseline agent and compare retrieval-augmented R2D2 (RA-R2D2) to vanilla R2D2. The retrieval dataset is the agent's current replay buffer. The agent process is parameterized as an GRU (Hochreiter & Schmidhuber, 1997), and the retrieval process is parameterized as the slot-based recurrent architecture described in Section 2, using 8 slots. We randomly sample a retrieval batch consisting of 256 trajectories from the retrieval dataset. In figure 2 we report the relative improvement of RA-R2D2 versus the R2D2 baseline. We observe an increase of $11.32 \pm 1.2\%$ in the mean human normalized score relative to the R2D2 baseline over 2 billion environment steps, demonstrating that retrieval augmentation is quite beneficial in Atari and

that the agent's own replay buffer is a useful source for retrieval. Raw scores and training curves are presented in appendix A.3.

### 3.1.1 ABLATIONS AND ANALYSIS

To understand the benefit of different components of retrieval augmentation, we ablate RA-R2D2 on the 10 Atari games it performs best relative to R2D2. The ablations are as follows.

*Importance of a separate retrieval process (A-1).* In R2A, the retrieval process and the agent process are parameterized separately, i.e., they have their own internal states. Here we examine what happens when the agent's state is used to query the retrieval batch instead of using the retrieval state $m_t$. To implement this we modify Step 1 of Algorithm 1 to make the query a direct function of the state of the agent, $q_t = f_{query}(s_t)$. The resulting query is used in the same way as above. This is similar to the episodic control baseline of (Pritzel et al., 2017).

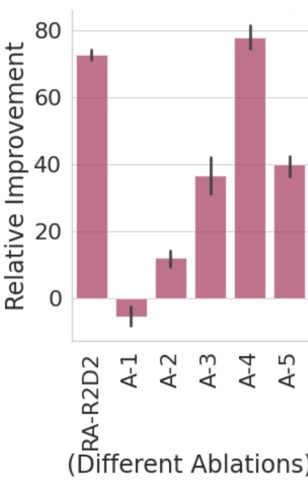

*Importance of retrieving information (A-2).* We examine what happens when the retrieval process does not have access to the retrieval dataset and hence no information is retrieved, keeping all else the same. In this ablation, the retrieval process updates the state of the slots using a transformer (i.e., in Step 1 we replace GRU with a transformer), and the updated state of the transformer is used by the agent process to shape the representation of its value function.

Figure 3: Relative performance of ablated RA-R2D2 versus baseline R2D2 for 5 ablations on 10 Atari games. black lines represent Standard deviation.

*Shorter retrieved trajectories (A-3).* We decrease the length of the trajectories that are retrieved and summarized during retrieval pre-processing, thus reducing the amount of past and future information the retrieval process can retrieve. By default, the trajectories in the retrieval dataset are of length 80. To perform this ablation, we decrease the length of the effective context to only include information from 5 timesteps.

*Importance of auxiliary losses to summarize retrieval batch (A-4, A-5).* Here we study the use of self-supervised BERT style masking losses in addition to using action, reward and value prediction. We use these auxiliary losses on top of the representation learned by the forward and the backward dynamics model. To implement these losses, we randomly mask 15% of the hidden states in a trajectory, and then using the representation of hidden states at other time-steps, we predict the representation of masked hidden states. In *A-5*, we study using *only* self-supervised BERT style masking losses for summarizing the trajectories.

Figure 3 shows the performance of RA-R2D2 and each ablation relative to the R2D2 baseline. Ablation A-1 shows that it is crucial to parameterize the agent process and retrieval process separately, as using the agent state does no better than the baseline. Ablation A-2 shows that the retrieval process finds relevant information that is useful to the agent. It also shows that the performance gains of R2A are not simply due to an increase in model capacity but depend on retrieving information from the retrieval dataset. Ablation A-3 shows that decreasing the length of the context in the retrieval dataset results in worse performance, thus showing the importance of our forward and backward trajectory summarization. Ablation A-4 demonstrates that the performance of R2A can further be improved by incorporating BERT style auxiliary losses but that only using BERT style auxiliary losses results in worse performance (but still better than the baseline R2D2). For completeness, Figure 10 in the appendix repeats the ablations on the 5 games that RA-R2D2 performs worst (relative to R2D2). The main takeaway is that the performance of the R2A can be greatly improved by optimizing hyperparameters for each game separately (A-6), which we did not do for our experiments.

### 3.2 GRIDROBOMAN: MULTI-TASK OFFLINE RL WITH A TASK-SPECIFIC RETRIEVAL DATASET

Beyond querying the agent's own experiences, retrieval can provide helpful information from other sources of experiences, including experts or other agents as the case in offline RL where the agent must learn from a fixed dataset of experiences generated by other agents without interacting with the environment during training. A major challenge in offline RL is distributional shift—the mismatch

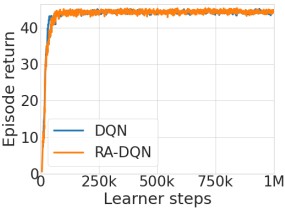 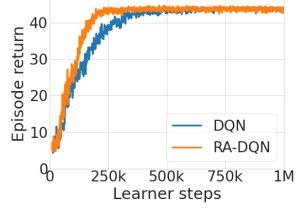 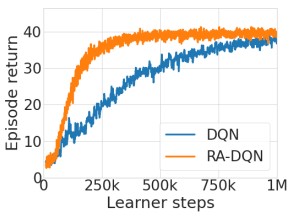

(a) Training and testing on 10 tasks.     (b) Training and testing on 20 tasks.     (c) Training and testing on 30 tasks.

Figure 4: **Gridroboman: Multi-task offline RL with a task-specific retrieval dataset.** Average episode return vs. learner steps for the multi-task gridroboman environment when training and evaluating on 10, 20, and 30 tasks. With fewer tasks (a), the baseline DQN agent (blue) and the retrieval-augmented DQN agent (orange) perform identically; however, when the number of tasks increases (b, c), the retrieval-augmented agent learns much more effectively than the baseline DQN agent. Results are the average of 3 seeds for each method. Curves for individual tasks are shown in the appendix (Figure 12).

between the distribution of states in the training data and those visited by the agent when acting—which makes it difficult to learn an accurate value function for states and actions rarely seen during training. We hypothesize that the retrieval process can improve performance in the offline setting by retrieving trajectories (including states, actions, and rewards) relevant to the agent's current state, particularly for states and actions that are rare in the offline dataset. We test this hypothesis on a multi-task offline RL setup where a single agent is trained on multiple tasks simultaneously but at training and evaluation time the retrieval dataset contains only trajectories from the evaluated task.

For this experiment, we created a minimalistic grid-world-based robotic manipulation environment (gridroboman) with 30 tasks related to the three objects (red, green, and blue) on the board. Gridroboman is built on the pycolab game engine (Stepleton et al.). The environment is inspired by the challenges of robotic manipulation, and includes tasks such as "go to object X" and "put object X on object Y". Details of the environment, its tasks, and an example figure are presented in the appendix (section A.4). Here, we incorporate retrieval augmentation into a vanilla DQN agent as agent-state-recurrence is not needed for this task. Figure 4 shows the results of training retrieval-augmented DQN (RA-DQN, orange) and DQN (blue) on increasing numbers of tasks. With fewer tasks, RA-DQN and DQN perform identically; however, when the number of training tasks increases the retrieval-augmented agent is able to learn much more effectively than the baseline agent. Training on more tasks requires either additional model capacity or the ability to extract information from fewer relevant samples for each task. By directly querying task-relevant experiences in the offline dataset, retrieval augmentation improves sample efficiency. Note that while the retrieval process does afford extra model capacity to the agent directly, ablation A-2 in section 3.1.1 shows that the retrieved information is what is crucial to performance, not the increased capacity.

### 3.3 BABYAI: MULTI-TASK OFFLINE RL WITH A MULTI-TASK RETRIEVAL DATASET

In this third experiment, our goal is to evaluate the benefit of retrieval augmentation when data from other tasks is present in the retrieval dataset. Multi-task retrieval data can be either harmful if the retrieved information misguides the agent or beneficial if information from the other tasks is relevant to the current task. Due to the use of attention in the retrieval process, we hypothesize that R2A will be able to ignore distracting information and retrieve relevant information from other tasks.

To test our hypothesis, we use the BabyAI environment (Chevalier-Boisvert et al., 2018), a partially observable multi-room grid world in which harder tasks are composed of simpler tasks and are formulated using subsets of a synthetic language. At the start of each episode, the agent is placed in a random room and must navigate to a randomly located goal. Due to the partial observability, we use a recurrent DQN (RDQN) agent as the baseline and compare its performance to a retrieval-augmented RDQN (RA-RDQN) agent. As is common in this environment, we measure the success rate of each agent, defined as the ratio of tasks the agent was able to accomplish given a fixed number of steps for each task. Table 1 shows the performance of RA-RDQN with a multi-task replay, RA-RDQN with a replay specific to the current task, and the baseline for varying amounts of offline training data (50K trajectories per task versus 200K trajectories per task). As expected from the previous experiment, retrieval augmentation improves performance over the baseline when using a single-task replay. Performance further improves when using a multi-task replay. We believe that this is due to

Table 1: **BabyAI: Multi-task offline RL with a multi-task retrieval dataset.** Mean success rate of retrieval-augmented recurrent DQN (RA-RDQN) as compared to a recurrent DQN (RDQN) baseline on the 40 BabyAI levels as function of the amount of training data. RA-RDQN is run twice, once with only the current task being evaluated in the retrieval dataset and once with all tasks in the retrieval dataset. Results are the average of 3 random seeds with standard errors.

| Method | Success Rate (50K) | Success Rate (200K) |
|---|---|---|
| RDQN | $32\% \pm 4\%$ | $45\% \pm 6\%$ |
| RA-RDQN (single-task retrieval buffer) | $48\% \pm 4\%$ | $64\% \pm 5\%$ |
| RA-RDQN (multi-task retrieval buffer, without IB) | $47\% \pm 3\%$ | $59\% \pm 6\%$ |
| RA-RDQN (multi-task retrieval buffer) | $55\% \pm 5\%$ | $74\% \pm 3\%$ |

the compositional nature of tasks in BabyAI, where retrieving information about the current subtask is more informative than retrieving information about the overall task.

## 4 RELATED WORK

**Episodic control.** The idea of allowing deep RL agents to adapt based on past experiences using a non-parametric memory is not new (Blundell et al., 2016; Pritzel et al., 2017; Hansen et al., 2018; Eysenbach et al., 2019; van Hasselt et al., 2019). The basic idea is that the agent is equipped with an episodic memory system, which is used to remember and recall past experiences to inform decisions. There are two important differences between R2A and these methods. (1) In these methods, a local action-value function is constructed by using information about the nearest neighbors in the replay buffer, and then the agent makes a decision about which action to execute based on both the local value function as well as the global value function. However, in the proposed work, we employ a parameterized network (the retrieval process), which has access to the information in the replay buffer, and the agent process uses the retrieved information to shape the predictions of its value function in a fully differentiable way (using attention). (2) In these episodic control methods, there is only one process (the agent), which has direct access to the replay buffer. However, in R2A, the agent process only has indirect access to the replay buffer via the retrieval process.

**Separation of concerns.** In Hierarchical RL (HRL) (Heess et al., 2016; Frans et al., 2017; Vezhnevets et al., 2017; Florensa et al., 2017; Hausman et al., 2018; Goyal et al., 2019c), there's separation of concerns among different policies, each policy focuses on a different aspect of the task, e.g., giving task relevant information to the high level policy only such that low level policy learns behaviours that are task agnostic. In these methods, the high level policy shapes the behaviour of low level policy by either influencing representations or by influencing rewards. It is possible to view our work through an analogous lens: wherein the "retrieval process" is the higher level policy (and has access to the all the information in the replay buffer) and is influencing the representation of the agent process that is interacting with the environment. However, there are also notable differences in our work—for instance, the agent process also directly shapes the representation of the retrieval process, which is generally not the case in HRL (e.g., in Vezhnevets et al. (2017) the manager directly influences the worker, but the worker does not directly influence the manager).

**Retrieval in language models.** Several retrieval-based methods have recently been developed for question answering, controllable generation, and machine translation (Guu et al., 2020; Lee et al., 2019; Lewis et al., 2020; Sun et al., 2021). The general scheme in such methods is to combine a parametric model (like a BERT-style masked language model or a pre-trained seq2seq model) with a non-parametric retrieval system. They show that such models generate diverse and factually correct language than the state of the art parametric only baselines. These methods share some similarities with our proposed model since they all involve a retrieval component but focus on different domains.

Additional related work is discussed in the appendix (section A.1).

## 5 CONCLUSION.

In this work, we developed R2A, an algorithm that augments an RL agent with a retrieval process. The retrieval process and the agent have separate states and shape the representation and predictions of each other via attention. The goal of the retrieval process is to retrieve useful information from a dataset of experiences to help the agent achieve its objective more efficiently and effectively. We show that R2A improves sample efficiency over R2D2, a strong off-policy agent, and compensates for insufficient capacity when training in a multi-task offline RL environments. Multiple ablations show the importance of the different components of R2A, including retrieving information from past

experiences and parameterizing the agent and retrieval process separately instead of giving the agent process direct access to the replay buffer.

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

# A   APPENDIX

## CONTENTS

## A.1   EXTENDED RELEVANT WORK.

**Model-based RL.** Their are different ways to integrate knowledge across past experiences. One of the most common way is by learning a model of the world, and using the predictions from the model to improve the policy and the value function (Sutton, 1991; Silver et al., 2008; Silver, 2009; Allen & Koomen, 1983; Silver et al., 2016; Racanière et al., 2017; Silver et al., 2018; Springenberg et al., 2020; Schrittwieser et al., 2020). To integrate information across different episodes (potentially separated by many time-steps), a model may needed be unrolled for many time-steps leading to compounding errors. In R2A, the agent has direct access to the information in the retrieval dataset, and querying across multiple trajectories (in parallel) in the retrieval dataset potentially separated by hundreds of time-steps.

**Structural Inductive Biases.** Deep learning have proposed structural inductive biases such as Transformers (Vaswani et al., 2017; Dehghani et al., 2018; Radford et al., 2019; Chen et al., 2020a;b; Dosovitskiy et al., 2020) or slot based recurrent architectures (Battaglia et al., 2016; Zambaldi et al., 2018; Battaglia et al., 2018; Goyal et al., 2019b; Watters et al., 2019; Goyal et al., 2020b; Veerapaneni et al., 2020; Goyal et al., 2021) where the induced structure has improved generalization, model-size scaling, and longrange dependencies.

**Reinforcement Learning with Offline Datasets.** Recent work in RL has tried exploiting large datasets collected across many tasks to improve the sample efficiency of RL algorithms (Vecerik et al., 2017; Pertsch et al., 2020; Nair et al., 2020; Siegel et al., 2020). An advantage of such large datasets is that they can be collected cheaply, and can then be reused for learning many downstream tasks. A general scheme for exploiting information about such task-agnostic datasets is either using them to directly improve the value function, or by extracting a set of skills or options and learning new tasks by recombining them. In our work, we try to use information in the replay buffer by querying and searching for the relevant information across multiple trajectories which otherwise would take many replays through coincidentally relevant information for this to occur.

**Efficient Credit assignment.** Learning long term dependencies requires assigning credit to time-steps far back in the past. Common methods for assigning credit in dynamics model like backpropagation through time requires information to be propagated backwards through every single step in the

past. This could become computationally expensive when used with very long sequences. Methods which try to get around this problem only back-propagate information through a selected time-steps in the past, realized by a learned mechanism that associates current state with relevant past states (Ke et al., 2018; Wayne et al., 2018; Arjona-Medina et al., 2018; Goyal et al., 2018; Fortunato et al., 2019). Most of these works consider assigning credit to states within the same trajectory, whereas the proposed model R2A, searches for the relevant information in the replay buffer which includes information from other trajectories also.

## A.2 Information Theoretic Formulation

Let's denote the agent's policy $\pi_\theta(A \mid S, G)$, where $S$ is the agent's state, $A$ the agent's action, $G$ the information from the retrieval process conditioned on the current state of the agent i.e., $G = f_{\text{retrieval}}(S)$, and $\theta$ the parameters of the neural network representing the policy. We want to train agents that in addition to maximizing reward, minimize the policy dependence on the information from the retrieval process, quantified by the conditional mutual information $I(A; G \mid S)$.

This approach of minimizing the dependence of the policy on the information from the retrieval process can be interpreted as encouraging agents to learn useful behaviours and to follow those behaviours closely, except where diverting from doing so (as a result of using information from the retrieval process) leads to higher reward (Strouse et al., 2018; Teh et al., 2017; Goyal et al., 2019a; Galashov et al., 2019; Czarnecki et al., 2019). To see this, note that the conditional mutual information can also be written as $I(A; G \mid S) = \mathbb{E}_{\pi_\theta}[D_{\text{KL}}[\pi_\theta(A \mid S, G) \mid \pi_0(A \mid S)]]$ where $\mathbb{E}_{\pi_\theta}$ denotes an expectation over trajectories, $\pi_0(A \mid S) = \sum_g p(g)\,\pi_\theta(A \mid S, g)$. is a "default" policy with the information from the retrieval process marginalized out. We maximize the following objective:

$$
\begin{aligned}
J(\theta) &\equiv \mathbb{E}_{\pi_\theta}[r] - \beta I(A; G \mid S) \\
&= \mathbb{E}_{\pi_\theta}[r - \beta D_{\text{KL}}[\pi_\theta(A \mid S, G) \mid \pi_0(A \mid S)]],
\end{aligned}
\tag{1}
$$

where $\beta > 0$ is a tradeoff parameter, $\mathbb{E}_{\pi_\theta}$ denotes an expectation over trajectories (for ex. generated by the agent's policy) and $D_{\text{KL}}$ refers to the Kuhlback-Leibler divergence.

(Goyal et al., 2019a; Galashov et al., 2019) proposes to optimize the Eq. 1 by maximizing the lower bound $\tilde{J}(\theta)$ [3]:

$$
J(\theta) \geq \tilde{J}(\theta) \equiv \mathbb{E}_{\pi_\theta}[r - \beta D_{\text{KL}}[p_{\text{enc}}(Z \mid S, G) \mid q(Z \mid S)]].
\tag{2}
$$

We parameterize the policy $\pi_\theta(A \mid S, G)$ using an encoder $p_{\text{enc}}(Z \mid S, G)$, a decoder $p_{\text{dec}}(A \mid S, Z)$ and the $q(Z \mid S)$ is the learned prior such that $\pi_\theta(A \mid S, G) = \sum_z p_{\text{enc}}(z \mid S, G)\, p_{\text{dec}}(A \mid S, z)$. [4] The encoder output $Z$ is meant to represent the information from the retrieval process that the agent believes is important to access in the present state $S$ in order to perform well.

Due to the data processing inequality (DPI) (Cover, 1999) $I(Z; G \mid S) \geq I(A; G \mid S)$, and hence to obtain an upper bound on $I(Z; G|S)$, we must first obtain an upper bound on $I(Z; G|S = s)$, and then average over $p(s)$. We get the following result:

$$
I(Z; G|S) \leq \sum_s p(s) \sum_g p(g|s) D_{\text{KL}}(p(Z|s, g) \| r(Z))
\tag{3}
$$

Such an information bottleneck has shown to improve generalization (Teh et al., 2017; Goyal et al., 2019a;c; Galashov et al., 2019; Merel et al., 2019; Liu et al., 2019; Goyal et al., 2020a; Liu et al., 2021).

## A.3 Atari: Implementation details and raw scores for R2D2 and RA-R2D2.

Table 3 shows the raw scores achieved by RA-R2D2 and the R2D2 baseline. For R2D2 baseline and for the parameterization of the agent process, we follow the exact same training setup as in (Banino et al., 2021). Figure 7, 8, 9, shows the learning curve for both the proposed model and the R2D2 baseline for different games.

---

[3] We ask the reader to refer to Information Theoretic Formulation section in the appendix.

[4] For experiments, we estimate the the marginals and conditionals using a single sample throughout.

Training Losses:

- The parameters of the state encoder are trained by the RL loss and the auxiliary losses used to train the forward/backward summarizer. The state encoder encodes information about the past actions, past rewards and the current observation into an abstract state.

- The parameters of the retrieval process are trained by the RL loss and the information bottleneck regularizer.

- The parameters of the agent process are trained only by the RL loss.

Hyperparameters:

- At each learner step, the agent samples a large batch of $512$ trajectories from the replay buffer. A fixed fraction of the batch ($64$ trajectories) is used for learning the Q-function and the remaining trajectories forms the retrieval batch.

- We sample a different retrieval dataset for each gradient update. The re-encoding of the trajectories is performed for each gradient update during training.

- The retrieval process selects the top-$k_{\text{traj}} = 10$ most relevant trajectories (in step 2, section 2.3), and then retrieves relevant information from the selected trajectories (step 3, section 2.3) using the top-$k_{\text{states}} = 10$ most relevant states (see 5).

- To summarize the experiences in the retrieval batch we use a forward and backward GRU with $512$ hidden units.

- We use 8 slots for Atari-R2D2 experiments.

- We use auxiliary losses in the form of action or policy prediction (section 2.2). We follow the same setup as proposed in (Schrittwieser et al., 2020).

- The value of the $\beta$ coefficient for the information bottleneck regularizer is fixed to $0.1$.

- The rest of the hyperparameters for R2D2 are taken from Kapturowski et al. (2018) and are detailed in table 2

| Hyperparameter | Value |
|---|---|
| Optimizer | Adam |
| Learning rate | 0.0001 |
| Q's $\lambda$ | 0.8 |
| Adam epsilon | $10^{-7}$ |
| Adam beta1 | 0.9 |
| Adam beta2 | 0.999 |
| Adam clip norm | 40 |
| Q-value transform | $h(x) = \text{sign}(x)(\sqrt{|x| + 1} - 1) + \epsilon x$ |
| Discount factor | 0.997 |
| Trace length (Atari) | 80 |
| Replay period (Atari) | 40 |
| Replay capacity | 100000 sequences |
| Replay priority exponent | 0.9 |
| Importance sampling exponent | 0.6 |
| Minimum sequences to start replay | 5000 |
| Target Q-network update period | 400 |
| Evaluation $\epsilon$ | 0.01 |
| Target $\epsilon$ | 0.01 |

Table 2: Hyperparameters used in the Atari R2D2 experiments.

Implementation details:

- We use GRU style gating and layernorm whenever we are changing the state of the agent process or the state of the slots in a residual way (Step 5, 6 in Algorithm 1).

- If the input to a neural network $f_{nn}$ consists of two tensors $x, y$, we never concatenate the inputs to the neural network. We make use of attention. We assume one of the inputs is the primary input (lets say $x$), and other input is privileged input (as a result of doing some expensive computation, let's say $y$). First, we compute $f_{nn}(x)$, and then using the output of this computation, we cross-attend over $y$ using multi-head attention.

- When computing the forward summary of the trajectories in the retrieval batch $h = h_t + s_t$, we also add the information about the current encoded state i.e., $s_t$.

**Visual Encoder** Visual observations are encoded using a ResNet-47 encoder. The 47 layers are divided in 4 groups which have the following characteristics:

- An initial stride-2 convolution with filter size 3x3 ($1 \cdot 4$ layers).

- Number of residual bottleneck blocks (in order): $(2, 4, 6, 2)$. Each block has 3 convolutional layers with ReLU activations, with filter sizes 1x1, 3x3, and 1x1 respectively ($(2+4+6+2)\cdot 3$ layers).

- Number of channels for the last convolution in each block: $(64, 128, 256, 512)$.

- Number of channels for the non-last convolutions in each block: $(16, 32, 64, 128)$.

- Group norm is applied after each group, with a group size of $8$.

After this observation encoding step, a final layer with ReLU activations of sizes $512$ is applied.

**Computational Complexity.** Here we discuss the computational complexity of the R2A.

The following computations are happening at each step within a trajectory:

- A particular slot selects the relevant trajectories and relevant states at each time-step.

- We use an attention mechanism to select the set of relevant trajectories and relevant states. This process scales with the number of trajectories in the retrieval batch.

- The set of relevant trajectories and relevant states changes across different time-steps within a trajectories. This process is repeated for all the slots independently, but can be efficiently performed on GPUs/TPUs.

We can make following optimizations to reduce the computational complexity of the R2A.

- For Atari and Gridroboman experiments instead of using a learned attention mechanism to rank the trajectories (in step 2, section 2.3), we can also just use those trajectories which are of high return (Abdolmaleki et al., 2018) since we already know that the data in the retrieval dataset is specific to the task on which we are training the agent.

- Once we have selected relevant trajectories, we can further reduce the computational cost of selecting most relevant states (in the selected trajectories) by only considering those states which are semantically similar to the agent's current state.

Improvements tried that seem to improve the performance but were not evaluated exhaustively:

- Instead of performing only single step of retrieval at each step, we can also query the replay buffer multiple times within a time-step at the cost of increased computational complexity.

- We can allow different slots to retrieve information at different time scales.

We note that all of the hyperparameters for the retrieval process, except for the number of trajectories in the retrieval batch, remain the same across all experiments (Atari, gridroboman, and BabyAI).

### A.3.1 RETRIEVAL PROCESS: RETRIEVING INFORMATION FROM THE RETRIEVAL BUFFER.

Here we discuss the different steps involved in retrieving information from the retrieval buffer:

**Query computation.** Each slot independently computes its prestate using the contextual information from the agent using a GRU. Then, each slot independently computes a *retrieval query* which will be matched against information in the retrieval buffer.

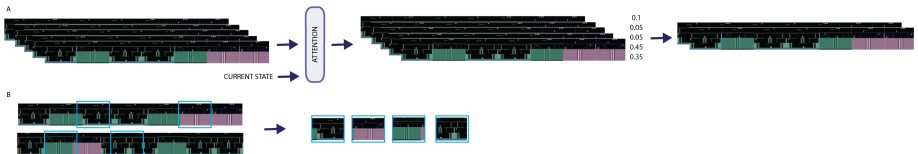

Figure 5: Retrieval of most relevant trajectories and states. A) The retrieval process selects the top-$k_{traj}$ most relevant trajectories as specified in step 2, section 2.3. In the figure scalars represent attention weights. B) Then the retrieval process retrieves relevant information from the selected trajectories by selecting the most relevant states from the top-$k$ trajectories as detailed in step 3, section 2.3

**Identification of most relevant trajectories and states for each slot.** The retrieval mechanism process uses an attention mechanism to match a query produced by the the retrieval state associated with each slot to keys computed on each time step of each trajectory of the retrieval batch. This process assigns a single scalar to each state within a trajectory. These scores are used to assign a scalar to each trajectory and then normalized across trajectories. These normalized scores are then used by the retrieval process to select the top-$k_{traj}$ most relevant trajectories (see Figure. 5). The process is then again repeated to select most relevant states with the selected trajectories. Here, a single scalar is assigned to each state with in selected trajectories and then normalized across all the states. These scores are then used to select most relevant states within the most relevant trajectories.

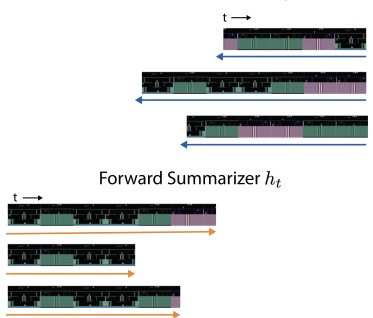

Figure 6: Each trajectory in the retrieval batch is summarized via a forward and a backward running parameteric structured model (Transformers or slot based recurrent network). Each state in the trajectory is represented by a $(h_t, b_t)$ tuple where $h_t$ represents the summary of the past and $b_t$ represents the summary of the future with that trajectory.

**Information retrieval from the most relevant states.** In the previous step, we selected most relevant states within the most relevant trajectories. Now we integrate information across the selected states. We again use the attention mechanism. Here the query by the slots is matched against the keys (which are the function of the forward running dynamics model), to retrieve information about the future (and values are the function of the backward running dynamics model). See figure. 6 for more details.

| Environment | R2D2 | RA-R2D2 |
|---|---|---|
| Alien | 52268 ± 4125 | **81626** ± 5123 |
| Amidar | 10976 ± 2341 | **24693** ± 1982 |
| Assault | **45521** ± 4751 | 43497 ± 6653 |
| Asterix | 997910 ± 51241 | 997585 ± 89251 |
| Asteroids | **262560** ± 94168 | 254194 ± 72141 |
| Atlantis | 1424723 ± 67589 | **1516463** ± 98140 |
| BankHeist | 3923 ± 512 | **19397** ± 5931 |
| BattleZone | **590913** ± 128715 | 466783 ± 89724 |
| BeamRider | **89038** ± | 79241 ± |
| Bowling | 250.37 ± | 253.5 ± |
| Boxing | 99 ± 0 | 99 ± 0 |
| Breakout | 848 ± 121 | **869** ± 51 |
| Centipede | 315329 ± 151612 | **497746** ± 12515 |
| ChopperCommand | 998842 ± 11451 | **999936** ± 3422 |
| CrazyClimber | 184132 ± 12912 | **226469** ± 9251 |
| Defender | 538752 ± 18241 | **554158** ± 21415 |
| DemonAttack | 143552 ± 0 | 143519 ± 0 |
| DoubleDunk | 24 ± 0 | 24 ± 0 |
| Enduro | 2332 ± 32 | 2359 ± 19 |
| FishingDerby | 71.357 ± 2.51 | 73.13 ± 3.12 |
| Freeway | 34 ± 0 | 33.19 ± 0 |
| Frostbite | 168225 ± 51891 | **511159** ± 124512 |
| Gopher | 123138 ± 4121 | 123841 ± 8914 |
| Gravitar | 13142 ± 1214 | 13198 ± 1102 |
| Hero | **43257** ± 4121 | 35431 ± 2412 |
| IceHockey | 72.74 ± 31 | 71.59 ± 12 |
| Jamesbond | 24873 ± 2141 | 25873 ± 2817 |
| Kangaroo | 14614 | 15232 |
| Krull | 158509 ± 21415 | **183921** ± 31415 |
| KungFuMaster | 208583 ± 2102 | **224385** ± 41512 |
| MontezumaRevenge | **1966.7** ± 987 | 1750 ± 1800 |
| MsPacman | 33530 ± 1214 | **44396** ± 4521 |
| NameThisGame | 30232 ± 3151 | **45140** ± 2412 |
| Phoenix | 695251 ± 32515 | **847914** ± 12415 |
| Pitfall | 0.0 ± 0 | 0.0 ± 0 |
| Pong | 21 ± 0 | 21 ± 0 |
| PrivateEye | **21602** ± 4124 | 18923 ± 2415 |
| Qbert | 242169 ± 51512 | 241525 ± 32513 |
| Riverraid | 32441 ± 5314 | **44554** ± 7325 |
| RoadRunner | 490872 ± 51512 | **513521** ± 65132 |
| Robotank | **128.44** ± 18 | 102 ± 29 |
| Seaquest | 998664 ± 1212 | **999899** ± 51 |
| Skiing | -29973 ± | 1212 **-29232** ± 2412 |
| Solaris | 3924 ± 3415 | 3853 ± 1241 |
| SpaceInvaders | 57198 ± 14125 | **62624** ± 4215 |
| StarGunner | 256129 ± | **478231** ± |
| Surround | 9.52 ± 0 | 10 ± 0 |
| Tennis | **7.16** ± 0 | 0 ± 0 |
| TimePilot | 168592 ± 21241 | **260609** ± 41212 |
| Tutankham | **390** ± 41 | 235 ± 89 |
| UpNDown | 544439 ± 5851 | **561389** ± 21516 |
| Venture | 2000 ± 0 | 2000 ± 0 |
| VideoPinball | 673679 ± | 234151 **962119** ± 24151 |
| WizardOfWor | 78431 ± 5159 | 73500 ± 6231 |
| Yars Revenge | **674692** ± 125116 | 513549 ± 89151 |
| Zaxxon | 53312 ± 15152 | **62631** ± 5161 |

Table 3: Scores obtained on different atari games. (average over 3 different seeds).

### A.3.2  DETAILS ON ABLATIONS

*Ablation A-1*: In this ablation, the internal retrieval state, $m_t$ is removed. The query $q_t$ is computed from $s_t$ directly as $q_t = f_{\text{query}}(s_t)$ . The retrieval vector $g_t$ is computed as per steps 2-4 of the original algorithm. Step 5 does not occur, and in step 6 the only update is $s_t \leftarrow s_t + g_t$.

*Ablation A-2*: In this ablation, there is no retrieval batch. The retrieval process is parametrized as before with a $n_k$-slot memory. Step 1 is identical, but steps 2-4 do not occur. Memory slots are updated as per the joint update of step 5. Step 6 is otherwise identical, except that keys and values are computed from the memory slots $m_t^k$ instead of the retrieved values $z_t^k$.

*Ablation A-3*: In this ablation, the length of the trajectories used for summarization is reduced from 80 to 5. The trajectories used for Q learning are unchanged.

*Ablations A-4 and A-5*: In our experiments, we use standard policy, reward and value prediction cross-entropy losses (Schrittwieser et al., 2020) to train the forward and backward summarizers. In A-4, we add a BERT loss; tokens are obtained by compressing the agent states $s_t$ with VQ-VAE. In A-5, we only use the BERT loss.

### A.3.3  ADDITIONAL ATARI ABLATIONS

Here we perform ablations on RA-R2D2 using the 5 Atari games on which RA-R2D2 performs worst relative to baseline R2D2. Figure. 10 shows the relative performance of different ablations compared to the R2D2 baseline. Ablations 1-5 are as described in the main text (section 3.1.1). For these results, we also ran a sixth ablation.

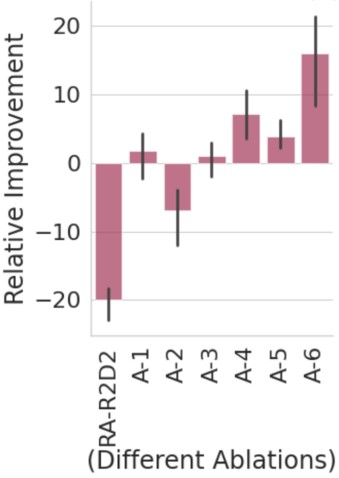

Figure 10: Relative performance of ablated RA-R2D2 versus baseline R2D2 for 6 ablations on 5 Atari games on which RA-R2D2 performs worse than baseline R2D2.

**Role of $k_{\text{traj}}$ and $k_{\text{states}}$ (A-6).** In our experiments, the retrieval process selects the top $k_{\text{traj}} = 10$ most relevant trajectories (step 2, section 2.3) and then selects the top $k_{\text{states}} = 10$ most relevant states of these trajectories (step 3, section 2.3) from which to retrieve relevant information. To better understand the role of these hyperparameters, we independently vary these two hyper-parameters (top $k_{\text{traj}}$ and top $k_{\text{states}}$) over the values $\{5, 10, 20\}$. Figure 10 shows that after independently varying these two hyper-parameters the performance of the R2A can be improved as compared to the R2D2 baseline.

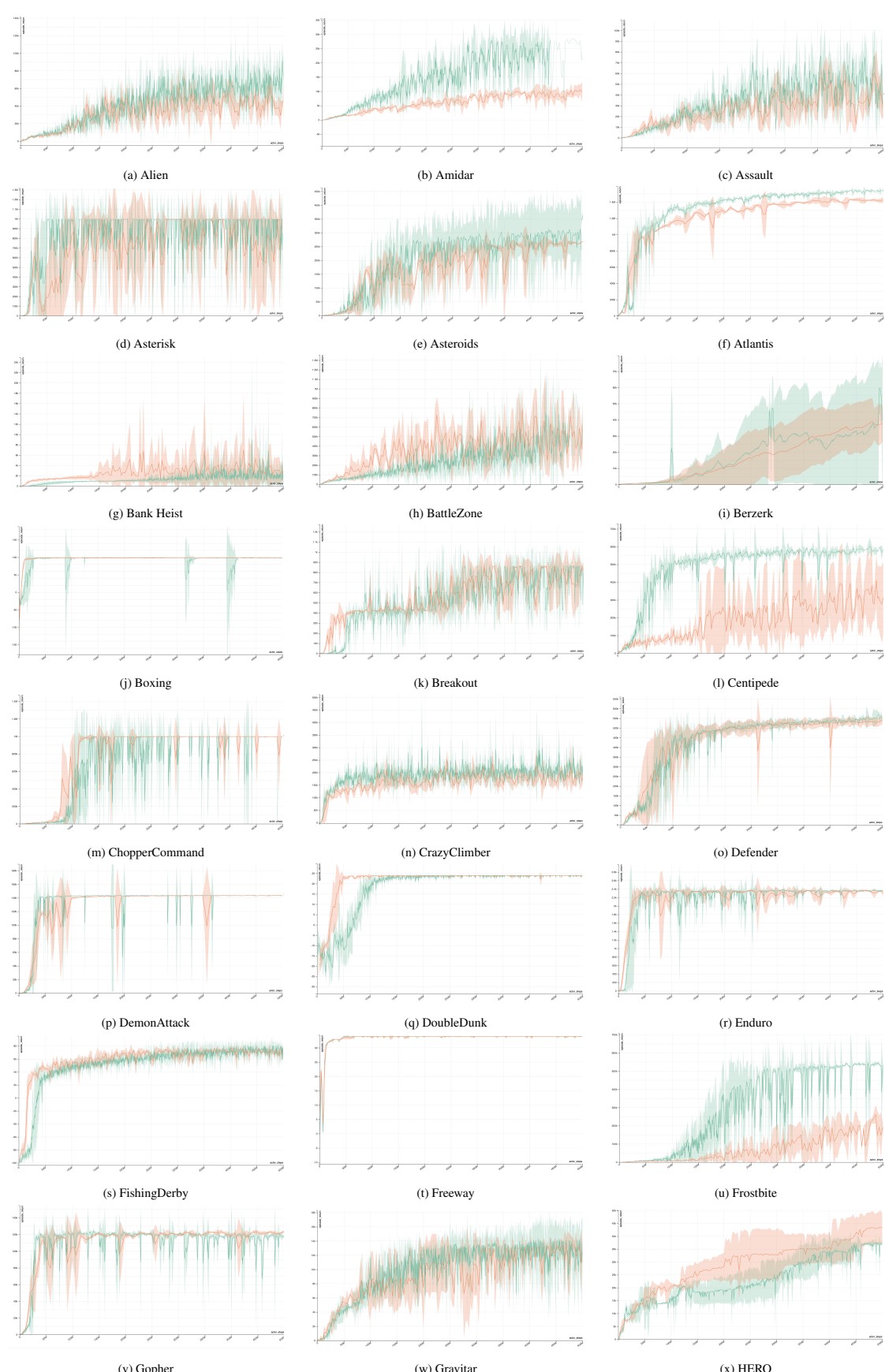

Figure 7: **Comparing R2A with R2D2:** Learning curves for retrieval-augmented R2D2 (RA-R2D2) and the R2D2 baseline across different Atari games. Green is RA-R2D2 and orange is baseline R2D2.

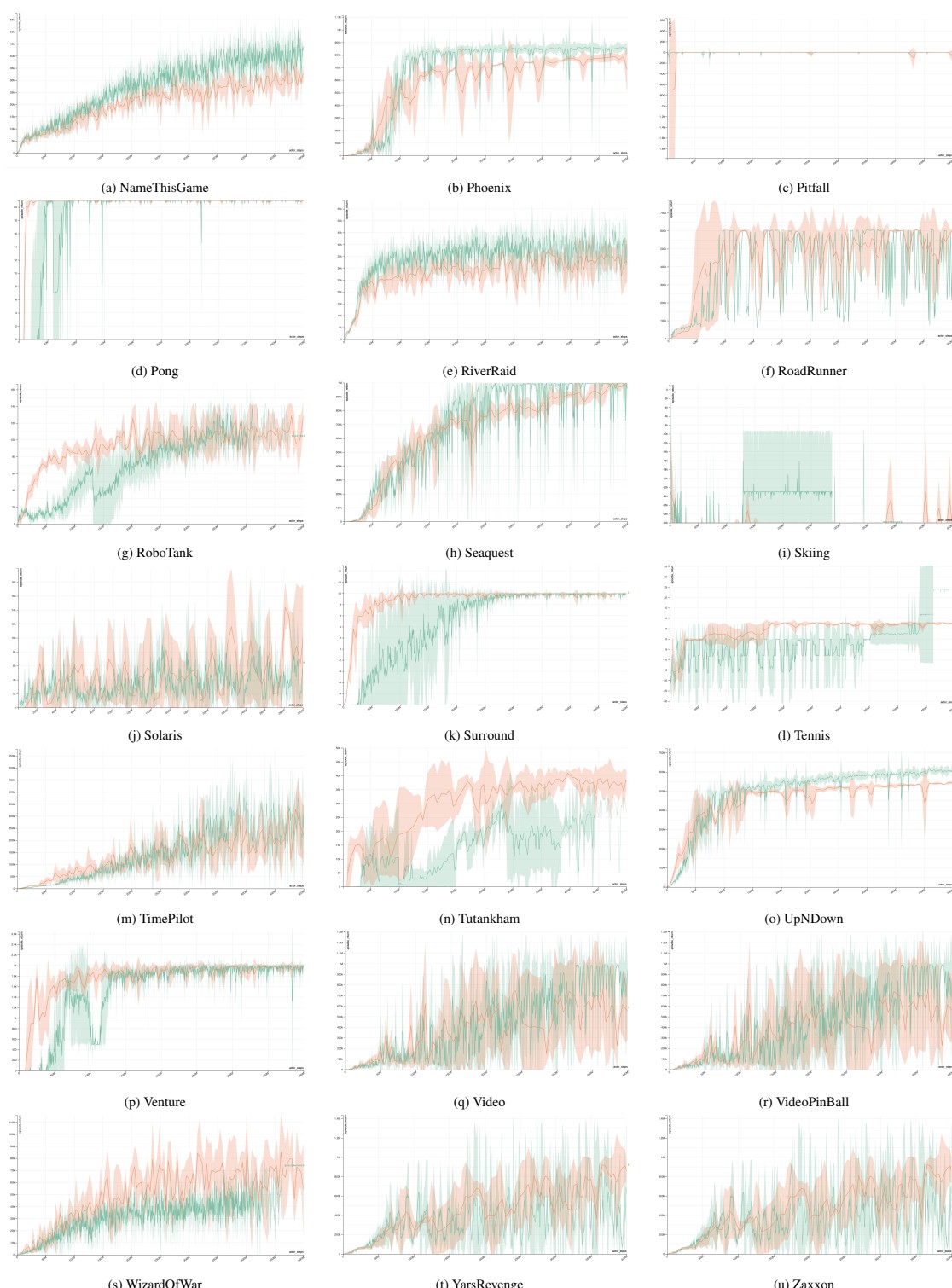

(a) NameThisGame (b) Phoenix (c) Pitfall
(d) Pong (e) RiverRaid (f) RoadRunner
(g) RoboTank (h) Seaquest (i) Skiing
(j) Solaris (k) Surround (l) Tennis
(m) TimePilot (n) Tutankham (o) UpNDown
(p) Venture (q) Video (r) VideoPinBall
(s) WizardOfWar (t) YarsRevenge (u) Zaxxon

Figure 8: **Comparing R2Awith R2D2:** Learning curves for retrieval-augmented R2D2 (RA-R2D2) and the R2D2 baseline across different Atari games. Green is RA-R2D2 and orange is baseline R2D2.

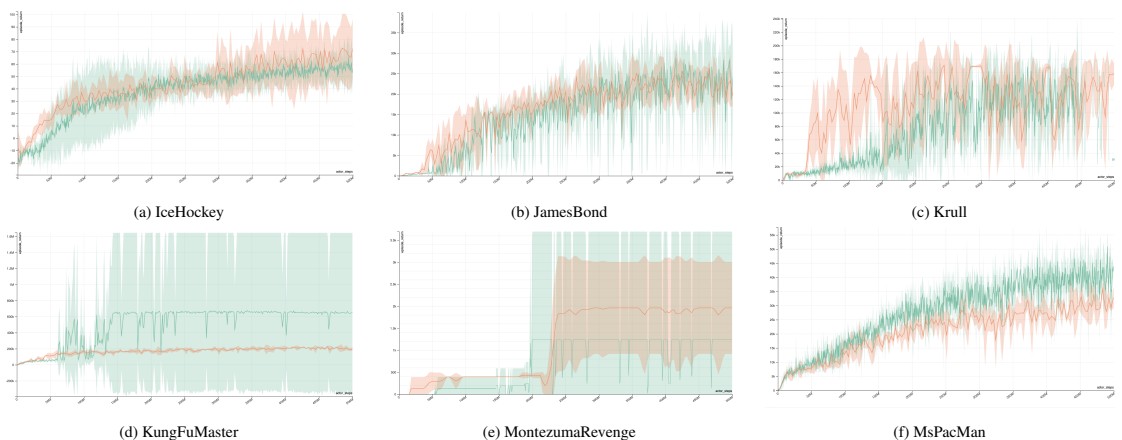

Figure 9: **Comparing R2Awith R2D2:** Learning curves for retrieval-augmented R2D2 (RA-R2D2) and the R2D2 baseline across different Atari games. Green is RA-R2D2 and orange is baseline R2D2.

## A.4 GRIDROBOMAN ENVIRONMENT

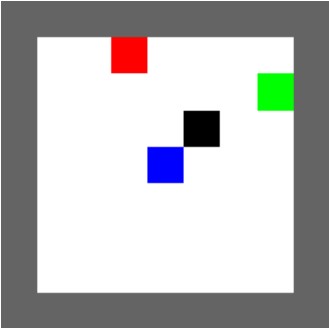

Figure 11: **Gridroboman environment illustration**. On the board there are three colored objects and the robot is represented by a black block. The robot can move itself and move the objects. The tasks are motivated by the robotic manipulation.

In order to test our method in a multi-task setting we designed a minimalistic grid world robotic manipulation (**gridroboman**) environment with a single embodiment and multiple tasks. It is implemented based on the pycolab (Stepleton et al.) game engine that provides tools for designing customizable grid world games. The environment and its tasks are inspired by the challenges in robotic manipulation. An illustration of a state in the environment is shown in Figure 4.

**Environment semantics, observations and actions** There are three colored objects on the $7 \times 7$ board: red, green and blue. The black block represents the robot. It can *move* in 4 directions and the action is skipped if it attempts to move into a wall (gray). Additionally, if the robot is located at the same position as any colored object, it can apply *lift* action that would enforce the object to move together with the robot. Then, *put* action allows to position the object either on the board or on the top of another single object. Additionally, there is an option of skipping an action. The initial state of the environment that includes positions of the objects and the robot is sampled randomly at the beginning of each episode. The agent observation is an 11-dimensional vector: it includes $x$ and $y$ coordinates of three objects and the robot as well as status of each of three objects. The status of an object is a numerical value: 0 if the object is positioned on the board, $-1$ if it is under another object and $+1$ if it is either held by the robot or located on the top of another object. Each episode lasts for 50 timesteps.

**Tasks** There are 30 possible tasks: lift red, green, or blue object (3), touch red, green, or blue object (3), move red, green, or blue to center (3) or to corner (3), touch one object with another (6), move two objects close to (3) or far from (3) each other, stack one object on the top of another (6). The nature of these tasks is such that it is impossible to identify the task by the initial state.

**Rewards** The reward is binary in each time step: it is zero when the task is not solved and one otherwise. In order to receive positive reward the following condition should be met:

- **lift X**: the robot should lift and not put back an object of color X;
- **touch X**: the robot should be located at the immediately adjacent cell to the object of X color and not hold any objects;
- **move X to center**: the object or color X should be located in the $3 \times 3$ block at the center of the board;
- **move X to corner**: the object of X color should be located in any of the $2 \times 2$ corners of the board;
- **touch X with Y**: the robot should be located at the immediately adjacent position to the object of X color and hold the object of color Y;
- **move X close to Y**: the distance between the objects of colors X and Y should be no more than 1 in both $x$ and $y$ direction;

- **move X far from Y**: the sum of distances between objects of colors X and Y in $x$ and $y$ coordinated should be greater than 9;

- **stack X on Y**: the objects of colors X and Y should be located at the same position one on the top of another.

### A.4.1 GRIDROBOMAN EXPERIMENT SETUP

The gridroboman offline RL dataset was generated by training a single DQN agent online on each task separately and recording the 100K generated episodes. The $Q(s, a)$ network was a 3-layer MLP with 256 units in each hidden layer for Q-function. For the experiments with 10, 20, and 30 tasks, we use the first 10, 20, and 30 tasks, as listed here.
**10 tasks:** touch red, touch green, touch blue, lift red, lift green, lift blue, red touch green, green touch red, and green touch blue.
**20 tasks:** The above 10 tasks, blue touch red, blue touch green, red to corner, green to corner, blue to corner, red to center, green to center, blue to center, red close to blue, and red close to green.
**30 tasks:** The above 20 tasks, blue close to green, red far from blue, red far from green, blue far from green, red on blue, red on green, green on red, green on blue, blue on red, and blue on green.

### A.4.2 GRIDROBOMAN HYPERPARAMETERS

The DQN agent used the same network as used to create the data: a 3-layer MLP with 256 hidden units per layer. The first 2 layers define $f_\theta^{\text{enc}}$ and the final layer predicts Q. The RA-DQN agent used the same base network plus a separate retrieval network.

Below we detail hyperparameters specific to gridroboman. Hyperparameters not specified in this section are the same as used in Atari.

- At each learner step, the agent samples a batch of 256 states from the replay buffer to train the DQN agent and further samples a batch of 64 trajectories from the retrieval dataset to form the retrieval batch.

- The retrieval process selects the top-$k_{\text{traj}} = 10$ trajectories with the highest return, and then retrieves relevant information from the selected trajectories (step 3, section 2.3) using the top-$k_{\text{states}} = 10$ most relevant states.

- To summarize the experiences in the retrieval batch we use a forward and backward GRU with 256 hidden units.

- We use 4 slots to parameterize the retrieval process.

- We use auxiliary losses in the form of value, reward, and policy prediction (section 2.2). We follow the same setup as proposed in (Schrittwieser et al., 2020) and weight these auxiliary losses with a coefficient of 0.1.

- The value of the $\beta$ coefficient for the information bottleneck regularizer is fixed to 0.3.

- The hyperparameters and setup for offline DQN are taken from Gulcehre et al. (2020). They are detailed in table 5. As in the above, we used double DQN (Van Hasselt et al., 2016).

| Hyperparameter | Value |
|---|---|
| Optimizer | Adam |
| Learning rate | 3e-4 |
| Discount factor | 0.99 |
| Importance sampling exponent | 0.2 |
| Minimum sequences to start replay | 5000 |
| TD loss function | Huber(1.0) |
| Target Q-network update period | 2500 |
| Evaluation $\epsilon$ | 6.5e-4 |

Table 4: Hyperparameters used in the gridroboman DQN experiments.

### A.4.3 GRIDROBOMAN EVALUATION CURVES WHEN TRAINING ON ALL 30 TASKS

Figure 12 we show the average episode reward obtained by the evaluation agent on each individual task when training on all 30 tasks. Curves are shown for the DQN and RA-DQN agents.

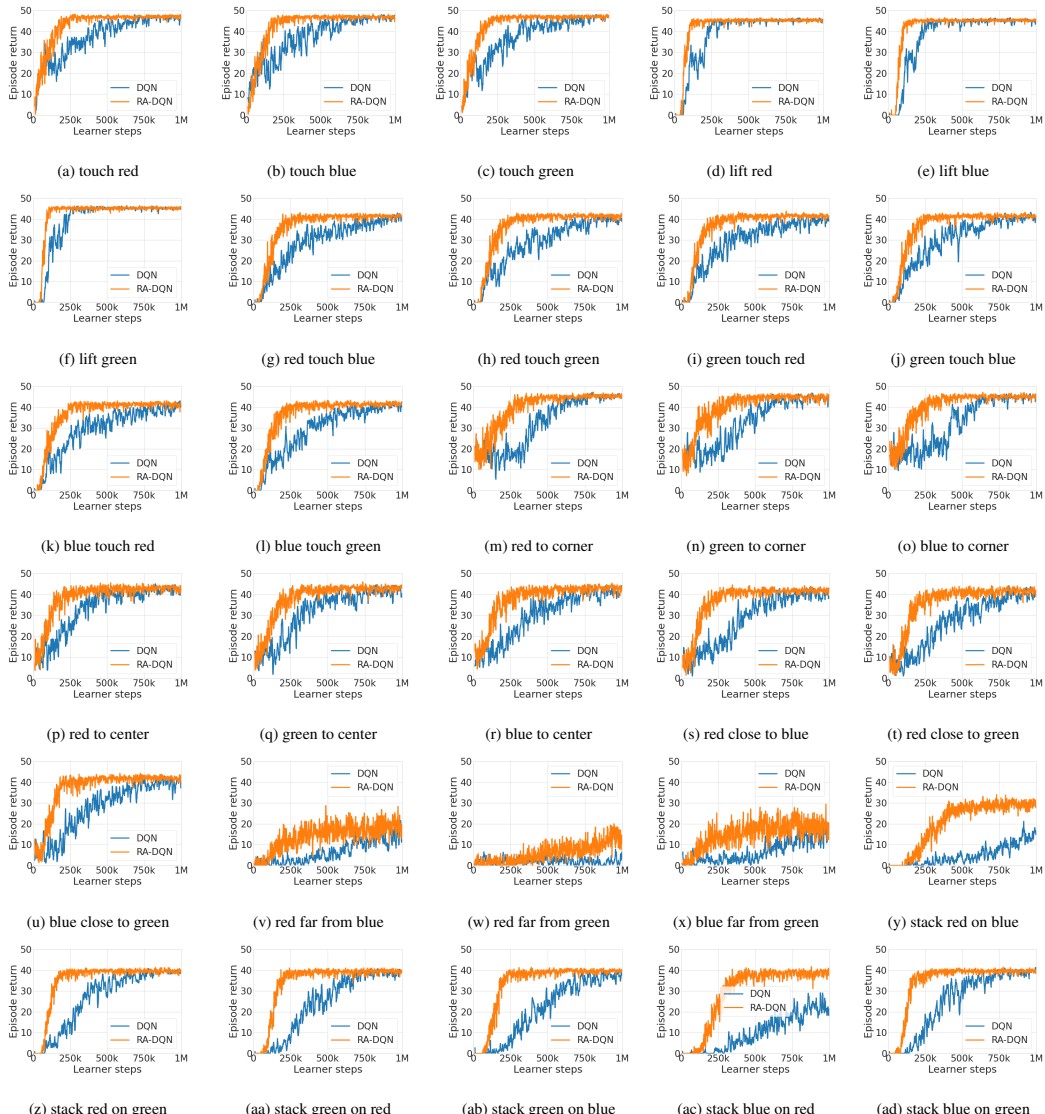

Figure 12: **Multi-task offline RL with a task-specific retrieval dataset.** Evaluation performance for RA-DQN (orange) and baseline DQN (blue) when training on all 30 gridroboman tasks with a single agent. Curves show the performance of each agent (averaged over 3 seeds) when running that agent online in the environment on the specified task.

## A.5 BABYAI ENVIRONMENT.

**Observations, actions.** We use the same setup for the observations and actions as in (Chevalier-Boisvert et al., 2018).

**Tasks.** We use all the 40 available tasks in the BabyAI environment. [5].

---

[5] https://github.com/mila-iqia/babyai/blob/master/docs/bonus_levels.md.
https://github.com/mila-iqia/babyai/blob/master/docs/iclr19_levels.md

**Hyper-parameters and RL algorithm.** We follow the same setup as in the Gridroboman experiments. Since BabyAI is a partially observable environment, we summarize the history of the agent using a recurrent encoder.

| Hyperparameter | Value |
|---|---|
| Optimizer | Adam |
| Learning rate | 3e-4 |
| Discount factor | 0.99 |
| Importance sampling exponent | 0.2 |
| TD loss function | Huber(1.0) |
| Target Q-network update period | 1000 |
| GRU hidden state | 512 |

Table 5: Hyper-parameters used in the BabyAI Recurrent DQN experiments.

**Retrieval Process.** We use the same hyper-parameters for the retrieval process as used in the gridroboman enviornment. For the case of multi-task retrieval buffer, we use 32 trajectories corresponding to each task to form the retrieval batch.

## A.6    LIMITATIONS AND FUTURE WORK.

It would be useful to investigate and extend the proposed idea in these different ways: (a) First, investigate training of the retrieval process and the agent process using different objectives as compared to training them in an end-to-end fashion, (b) Second, scaling R2A to more complex problems multi-agent problems like in Starcraft (Vinyals et al., 2019), where one may have access to millions of past experiences. In R2A, we only query a subset of the retrieval dataset, which limits the generality of the method. (c) Third, even more intriguing would be the possibility of learning an abstract model with abstract internal actions, and rewards, rather than learning a model which queries for information from the retrieval dataset, and hence avoiding the need for Monte Carlo tree search common in the state of the art planning methods (Schrittwieser et al., 2020). We aim to formulate these problems and seek answers in the future work.

## A.7    SOCIAL IMPACT.

The authors do not foresee negative social impact of this work beyond that which could arise from general improvements in ML.

## A.8    REBUTTAL PHASE.

**Comparison to Episodic Control baselines.**    We have performed ablations where we compared to the scenario where we just keep the simple episodic memory (refer to section. 3.1.1, ablation A-1). The result of ablations shows that for R2D2 accessing the dataset in a non-parametric way actually hurts the performance of the RL algorithm.

**BERT style unsupervised losses then not used generally in the R2A method**    BERT style unsupervised losses actually improve the learned representations. We did not choose to use BERT style representations as they increase the number of design choices, increase the number of ablations, increase the number of parameters, whereas the relative gains of using BERT style unsupervised losses is not much. Future work should investigate the use of pre-trained encoders as common in NLP literature.

**For Frostbite, episodic control is beneficial.**    As mentioned in "Building Machines That Learn and Think Like People", in Frostbite, players control an agent (Frostbite Bailey) tasked with constructing an igloo within a time limit. The igloo is built piece-by-piece as the agent jumps on ice floes in water The challenge is that the ice floes are in constant motion (moving either left or right), and ice floes only contribute to the construction of the igloo if they are visited in an active state (white rather than blue). The agent may also earn extra points by gathering fish while avoiding a number of fatal hazards (falling in the water, snow geese, polar bears, etc.). Success in this game requires a

temporally extended plan to ensure the agent can accomplish a sub-goal (such as reaching an icefloe) and then safely proceed to the next sub-goal. Ultimately, once all of the pieces of the igloo are in place, the agent must proceed to the igloo and thus complete the level before time expires. One hypothesis for the better performance could be that the retrieval process is able to efficient utilize the information from the states which are very far apart from the current state (i.e., temporally extended credit assignment). This hypothesis is further validated when we decrease the length of the traces in the dataset. In ablation section, if we decrease the length of the trajectories that are retrieved and summarized during retrieval preprocessing, thus reducing the amount of past and future information the retrieval process can retrieve. By default, the trajectories in the retrieval dataset are of length 80. To perform this ablation, we decrease the length of the effective context to only include information from 5 time-steps.

**Similar Computation baseline.**    We also compared to a baseline which does the same amount of "computations" (Importance of retrieving information (A-2)). We examine what happens when the retrieval process does not have access to the retrieval dataset and hence no information is retrieved, keeping all else the same. In this ablation, the retrieval process updates the state of the slots using a transformer (i.e., in Step 1 we replace GRU with a transformer), and the updated state of the transformer is used by the agent process to shape the representation of its value function. Since the use of the transformer number of parameters in this baseline is more as compared to R2D2. Our results show that the proposed method R2A archives better performance as compared to vanilla R2D2 as well as the baseline which does the same amount of computation (but does not retrieve information).

**How does runtime compare to vanilla R2D2?  How many more parameters are introduced?**
As compared to vanilla R2D2, a naive implementation of the proposed method is around 3x slow (and somewhat optimized implementation around 2x slower), with the increase of 10% more parameters. In the ablations, we also examine what happens when the retrieval process does not have access to the retrieval dataset and hence no information is retrieved but everything else is kept the same (same computation and same parameter baseline).

**Use of hierarchical attention (top-$k$ trajectories and then top-$k$ states).**    A hierarchical attention where we first filter trajectories, and then filter states reduces the search space for the attention. For example. consider 512 trajectories, and 80 states in each trajectory, then the attention operator has to select 10 most relevant states out of 40960 states.  Whereas if we do initial filtering of top-k trajectories, then the attention operator needs to select 10 most relevant trajectories out of 512 trajectories, and then 10 most relevant states out of 800 states. Such an hierarchical attention has also shown to be successful in various NLP tasks also.

**Analysis about retrieved information in the multi-task setting?**    We did some analysis to study the properties about the retrieved information in the multi-task setting in BabyAI. BabyAI setup contains about 40 tasks. Out of these 40 tasks, around 15 tasks are compositional i.e., requires to compose information from 2 or more sub-tasks, and rest of the tasks requires the agent to execute a particular behaviour (ex. going to the door, fetching a key etc).

We study as to how often the agent retrieves information from other tasks. Ideally, for the compositional tasks, the hope would be the agent access information about the other tasks more often as compared to the tasks which require only a particular behaviour. So, during test time, we study the percentage of times agent access information from the same task as compared to accessing information from the different tasks. For compositional tasks, the agent access information from other tasks about 54% of times, while the percentage for single tasks is about 21%. That said, we were not able to find any pattern as to find why the agent is retrieving more information from other tasks for some particular tasks as well as any structure about the states from which information is being retrieved.

