# OpenReview forum: "Retrieval-Augmented Reinforcement Learning"
_ICLR.cc/2022/Conference — ICLR 2022 Submitted_

### Official Review · Reviewer_LSKp · 2021-10-29

**Correctness:** 4
**Technical Novelty And Significance:** 3
**Empirical Novelty And Significance:** 4
**Recommendation:** 6
**Confidence:** 3

**Main Review:**

Strengths:
* The paper is well written, and despite the relative complexity of the R2A process I was able to understand most of the details with only a few additional passes.
* The use of an attention mechanism over the sampled replay buffer I believe is novel, and is a worthy addition to the episodic control literature
* There are a few promising results (e.g., BabyAI), and the ablations appear to show the importance of the introduced components.
* The creation of the `gridroboman` offline benchmark is a really nice addition, and should be commended.

Weaknesses:
* The experimental methodology is lacking in several aspects:
   * There should be significance testing, and there are quite a low number of seeds (3) across all experiments; ideally there should be a minimum of 5. This is further justified by the high variance of the training curves in the appendix.
   * The paper should compare to other episodic control approaches, such as [1,2]. This is because the idea of augmenting a controller with instantaneous buffer information is not new (i.e., episodic control), so I would like to see the merit to adopting a hierarchical + modern attention mechanism over say a neural dictionary approach.
  * There should be information about run time and computational resources compared to the vanilla R2D2 agent due to the additional machinery introduced by R2A.
  * More details about the BabyAI experiment, such as training curves and number of epochs; were the success rates reported when all agents had reached convergence? This is important as viewing the `gridroboman` results shows that although R2D2 learns slower offline, it does also eventually seem to reach the level of performance of R2A.
  * I'm somewhat concerned by the inclusion of the information bottleneck regularizer; is this necessary for the method to work generally? There should be an ablation with and without this.
* BERT style unsupervised loss improves the learned representations in Figure 3, but why is this then not used generally in the R2A method? Does it actually harm performance beyond the top 10 Atari games?
* In some cases, I'm not entirely convinced by the improvements offered by R2A over R2D2. Particular areas of concern are:
   * In more than 50% of all Atari games, there seems to be no improvement adding the retrieval mechanism. Furthermore the learning curves in the back don't convince me that overall R2A learns quicker either. This would be ok if it could be justified (e.g., is there something intrinsic to `Frostbite` that means episodic control is beneificial v.s. `Tutankham`?).
   * While R2A learns `gridroboman` quicker, in the offline setting this is less of a concern, since we are not exploring and so the only incurred cost is computational (not data-efficiency). Therefore the results here aren't that convincing to me, as it seems R2D2 can retrieve R2A performance in the limit; is this always the case? Furthermore, if R2A is more expensive to train per step, perhaps if time was placed on the x-axis the improvements of R2A would be not as clear?
* More details about the process of double filtering of top-k trajectories and states would be welcome. Is it not the case that we could be prefilterting out useful state information in the first round of top-10 trajectory information? (e.g., what if there was actually the most useful state information was in the *11th* most useful trajectory, but we dismissed it because on average it also contains some irrelevant information). Perhaps some experiments to show why we don't just perform attention over the entire set of retrieved states, (e.g., top-10 states over all 512 trajectories, not just top-10 trajectories).

Nits:
* LSTM and GRU appear to be used interchangeably, when they are different models, for instance in **Step 1: Query Computation** section of page 5.

Refs:

[1] Neural Episodic Control, Pritzel et al. arxiv:1703.01988

[2] Fast deep reinforcement learning using online adjustments from the past, Hansen et al., arxiv:1810.08163

**Summary Of The Paper:**

In this paper, the authors introduce R2A, an agent that is augmented with an attention-based retrieval mechanism that augments its state with information directly from an experience replay. They show that this mechanism can improve on R2D2 online performance, and can also improve learning in the offline domain.

**Summary Of The Review:**

Overall the paper presents a nice idea and is well written, and introduces a novel offline multi-task benchmark. However I believe there are shortcomings in the experimental methodology, and furthermore despite the relatively large increase in complexity introduced by R2A, I don't believe the authors have enough convincing the reader that this increase is worth it.

I would therefore recommend at this point a weak reject, but believe if the authors could address the points made concerning experimental methodology and design choice explanations, I'd be willing to increase this score.

---

> ### Author Response · Authors · 2021-11-16
> **Continuous Control Experiments + Episodic control baseline + Information Bottleneck  + BERT style unsupervised losses (1/2)**
>
> We thank the reviewer for taking time and reviewing our work. We appreciate the thorough feedback. We have run additional experiments to  incorporate reviewer's feedback. We believe the extra experiments address most of the concerns of the reviewer.
>
> **The paper should compare to other episodic control approaches, such as [1,2]**
>
> We have performed ablations where we compared to the scenario where we just keep the simple episodic memory (3.1.1 ABLATIONS AND ANALYSIS, ablation A-1). The result of ablations shows that for R2D2 accessing the dataset in a non-parametric way actually hurts the performance of the RL algorithm. We would update the description to make it more clear that these are episodic control baselines.
>
> **inclusion of the information bottleneck regularizer; is this necessary for the method to work generally?**
>
> Because we evaluate Atari for a billion environmental steps, it may not be possible to get the ablation results for atari within the discussion period. So we choose to run these ablations for the BabyAI experiments.
>
> RA-RDQN (multi-task retrieval buffer) with information bottleneck achieves 55% ± 5%, 74% ± 3% with 50K and 200K trajectories. Without information RA-RDQN (multi-task retrieval buffer) achieves 47\% ± 3\% and 59\% ± 6%. These results show the importance and the utility of the information bottleneck. We note that such information bottleneck regularizers have also shown to improve generalization as in [1,2].
> [1] Infobot,https://arxiv.org/abs/1901.10902, [2] Information asymmetry, https://arxiv.org/abs/1905.01240
>
> **New experiments on continuous control RL tasks CausalWorld**
>
> As Reviewer yAsc, suggested we have ran more experiments for continuous control RL tasks. We performed additional experiments on the CausalWorld dataset (https://arxiv.org/abs/2010.04296). We choose CausalWorld because it is a benchmark for causal structure and transfer learning in a robotic manipulation environment. The environment is a simulation of an open-source robotic platform, hence offering the possibility of sim-to-real transfer. Tasks consist of constructing 3D shapes from a given set of blocks - inspired by how children learn to build complex structures. The key strength of CausalWorld is that it provides a combinatorial family of such tasks with common causal structure and underlying factors (including, e.g., robot and object masses, colors, sizes). The user (or the agent) may intervene on all causal variables, which allows for fine-grained control over how similar different tasks (or task distributions) are. We use the following four tasks: (a) Pushing (b) Picking ( c ) Pick and Place (d) Stacking2. We follow the multi-task setting as mentioned in section 3.3.. Multi-task retrieval data can be either harmful if the retrieved information misguides the agent or beneficial if information from the other tasks is relevant to the current task. Since there’s some shared structure in the above selection of tasks, we hypothesize that R2A will be able to ignore distracting information and retrieve relevant information from other tasks. We followed the same experimental protocol as in CausalWorld, and used SAC as an RL algorithm. We train all the baselines and the proposed method for 100M environment steps in a multi-task fashion. We report the fractional success rate (i.e., as to how many times the agent is able to solve the task at hand).
>
> Multi-task SAC achieves a success rate of 73.5 +/- 3.6 %, RA-RSAC (single-task retrieval buffer) achieve a success rate of 82.5 +/- 5%, RA-RSAC (multi-task retrieval buffer) achieves a success rate of 86.4 +/- 4.3% (5 seeds).
>
> **why is BERT style unsupervised losses then not used generally in the R2A method?**
>
> The reviewer is right. BERT style unsupervised losses actually improve the learned representations. We did not choose to use BERT style representations as they increase the number of design choices, increase the number of ablations, increase the number of parameters, whereas the relative gains of using BERT style unsupervised losses is not much. Future work should investigate the use of pre-trained encoders as common in NLP literature.

---

> > ### Author Response · Authors · 2021-11-16
> > **Frostbite + computational cost + double filtering (2/2)**
> >
> > **something intrinsic to Frostbite that means episodic control is beneficial**
> >
> > As mentioned in “Building Machines That Learn and Think Like People”, in Frostbite, players control an agent (Frostbite Bailey) tasked with constructing an igloo within a time limit. The igloo is built piece-by-piece as the agent jumps on ice floes in water  The challenge is that the ice floes are in constant motion (moving either left or right), and ice floes only contribute to the construction of the igloo if they are visited in an active state (white rather than blue). The agent may also earn extra points by gathering fish while avoiding a number of fatal hazards (falling in the water, snow geese, polar bears, etc.). Success in this game requires a temporally extended plan to ensure the agent can accomplish a sub-goal (such as reaching an icefloe) and then safely proceed to the next sub-goal. Ultimately, once all of the pieces of the igloo are in place, the agent must proceed to the igloo and thus complete the level before time expires. One hypothesis for the better performance could be that the retrieval process is able to efficient utilize the information from the states which are very far apart from the current state (i.e., temporally extended credit assignment). This hypothesis is further validated when we decrease the length of the traces in the dataset. In section 3.1.1 ABLATIONS AND ANALYSIS, in ablation (Shorter retrieved trajectories (A-3)), we decrease the length of the trajectories that are retrieved and summarized during retrieval preprocessing, thus reducing the amount of past and future information the retrieval process can retrieve. By default, the trajectories in the retrieval dataset are of length 80. To perform this ablation, we decrease the length of the effective context to only include information from 5 timesteps.
> >
> > **While R2A learns grid roboman quicker, in the offline setting this is less of a concern, since we are not exploring and so the only incurred cost is computational (not data-efficiency).**
> >
> > We also compared a baseline which does the same amount of “computations” (Importance of retrieving information (A-2)) in section 3.1.1 ABLATIONS AND ANALYSIS. We examine what happens when the retrieval process does not have access to the retrieval dataset and hence no information is retrieved, keeping all else the same. In this ablation, the retrieval process updates the state of the slots using a transformer (i.e., in Step 1 we replace LSTM with a transformer), and the updated state of the transformer is used by the agent process to shape the representation of its value function. Since the use of the transformer number of parameters in this baseline is more as compared to R2D2. Our results show that the proposed method R2A archives better performance as compared to vanilla R2D2 as well as the baseline which does the same amount of computation (but does not retrieve information).
> >
> > Also for GridRoboman, If the number of tasks is less (20), the R2D2 can retrieve R2A performance in the limit, but if we increase the number of tasks, R2A performance is more as compared to R2D2 performance. Similarly, for BabyAI experiments at the end of training performance of RA-RDQN is strictly higher as compared to vanilla DQN (even in the limit of the training). Same is true for a subset of the Atari games.
> >
> > **More details about the process of double filtering of top-k trajectories and states would be welcome.**
> >
> > It could certainly be true. The hope is that when the process of top-k filtering is learned (again using an attention mechanism), the attention mechanism should be able to figure out what information is relevant. In appendix, A.3.3 ADDITIONAL ATARI ABLATIONS, where we perform ablations on RA-R2D2 using the 5 Atari games on which RA-R2D2 performs worst relative to baseline R2D2. Figure. 10 shows the relative performance of different ablations compared to the R2D2 baseline. Ablations 1-5 are as described in the main text (section 3.1.1). For these results, we also ran a sixth ablation where we vary the top_ktraj and top_kstates independently for each game. In our experiments, the retrieval process selects the top k_traj = 10 most relevant trajectories (step 2, section 2.3) and then selects the top k_states = 10 most relevant states of these trajectories (step 3, section 2.3) from which to retrieve relevant information. To better understand the role of these hyperparameters, we independently vary these two hyper-parameters (top k_traj and top k_states) over the values {5, 10, 20}. Figure 10 shows that after independently varying these two hyper-parameters the performance of the R2A can be improved as compared to the R2D2 baseline. So this resonates with what the reviewer is saying. Future work could investigate a dynamic way to choose the value of these hyper-parameters.

---

> > > ### Comment · Reviewer_LSKp · 2021-11-18
> > > **Thank you for your response**
> > >
> > > Thank you for your detailed response. It has gone some way in addressing my concerns, and I'm more comfortable with certain elements of the paper.
> > >
> > > I have a quick follow up: if Ablation A-1 is analogous to prior work on neural episodic control, why doesn't at least provide some benefit? For instance in the original [NEC paper](https://arxiv.org/pdf/1703.01988.pdf), their method clearly improves over their baseline; therefore I think it's unfair to say that ablation A-1, which reduces performance compared to R2D2, is a fair representation of prior work.
> > >
> > > I also still have a couple of questions that weren't answered:
> > >
> > > * How does runtime compare to vanilla R2D2? How many more parameters are introduced?
> > > * Why can't we just filter the top-k states in the entire set of 512 trajectories?

---

> > > > ### Author Response · Authors · 2021-11-19
> > > > **Further Clarifications**
> > > >
> > > > **It has gone some way in addressing my concerns, and I'm more comfortable with certain elements of the paper**
> > > >
> > > > We thank the reviewer for engaging with us. We feel motivated that our reply has helped to improve the clarity of the ideas presented in the paper.
> > > >
> > > > **why doesn't at least provide some benefit**
> > > >
> > > > In order to understand this question, let's try to first understand the results from NEC paper (https://arxiv.org/abs/1703.01988). If we look at Table 1 and Table 2 which summarizes the main results of the NEC paper, NEC seems to improve the sample efficiency in the low data regime (< 20 M time-steps). For 40M time-steps other baselines obtain much better results as compared to NEC/MFEC methods. Similarly, for the Atari results in the paper, we ran for 2B time-steps.  In the limit of 2B time-steps, NEC performs worse as compared to the R2D2.  This resonates with the results in the table 1 and table 2 where in the 40 M time-steps, NEC performs worse.
> > > >
> > > > **How does runtime compare to vanilla R2D2? How many more parameters are introduced**
> > > >
> > > > As compared to vanilla R2D2, a naive implementation of the proposed method is around 3x slow (and somewhat optimized implementation around 2x slower), with the increase of 10\% more parameters. In the ablations, we also examine what happens when the retrieval process does not have access to the retrieval dataset and hence no information is retrieved but everything else is kept the same (same computation and same parameter baseline). For more details, refer to section 3.1.1 ABLATIONS AND ANALYSIS, in ablation section.
> > > >
> > > > **Why can't we just filter the top-k states in the entire set of 512 trajectories?**
> > > >
> > > > A hierarchical attention where we first filter trajectories, and then filter states reduces the search space for the attention. For example. consider 512 trajectories, and 80 states in each trajectory, then the attention operator has to select 10 most relevant states out of 40960 states. Whereas if we do initial filtering of top-k trajectories, then the attention operator needs to select 10 most relevant trajectories out of 512 trajectories, and then 10 most relevant states out of 800 states. Such an hierarchical attention has also shown to be successful in various NLP tasks also.
> > > >
> > > > We don't have saved plots for BabyAI and it takes a week to get results (i.e., after the paper update deadline).

---

> > > > > ### Comment · Reviewer_LSKp · 2021-11-19
> > > > > **Thanks again**
> > > > >
> > > > > Great, thanks for the quick reply.
> > > > >
> > > > > Overall I think the paper is a good addition to the neural episodic control literature. The biggest drawback is the method's complexity, but the extensive ablations help clarify for me that each element appears to help final performance. The complexity also affects the issue of runtime (2-3x is quite significant), but given the novelty of this work (I believe it is the first that applies attention/transformer to episodic control), I think this can be forgiven somewhat. It is likely that certain elements could be redundant, but just getting attention to work in this setting should be commended, and future work can seek to make this more streamlined. Also the authors implementing R2A in SAC (and getting what appears to be good performance) helps assuage doubts this approach 'overfits' to R2D2.
> > > > >
> > > > > I'm still not entirely convinced by the performance; for instance looking at Fig. 4c, I'd argue that it's not true to say that vanilla R2D2 wouldn't recover the performance of R2A beyond 1m offline steps. Also I did not see updated results in the paper, nor training curves for BabyAI, which could reveal more about asymptotic performance than just  reporting the numbers in a table. Also for me a clear benefit of this approach appears to be in learning how to leverage multi-task experiences effectively, and the method appears strong in these settings. However some validation that indeed the mechanism is accessing these multi-task experiences would help further elucidate/prove this point for me. Perhaps some analysis could be done to see how often the top-k states don't come from the current task in the multi-task setting?
> > > > >
> > > > > With this in mind, I'm going to raise my score, but would need a bit more; I think an updated manuscript with these new elements would convince me to raise more.

---

> > > > > > ### Author Response · Authors · 2021-11-19
> > > > > > **Analysis about the multi-task BabyAI setup**
> > > > > >
> > > > > > We thank the reviewer for increasing their score. We appreciate it.
> > > > > >
> > > > > > **Perhaps some analysis could be done to see how often the top-k states don't come from the current task in the multi-task setting?**
> > > > > >
> > > > > > As reviewer asked, we did some analysis to study the properties about the retrieved information in the multi-task setting in BabyAI.
> > > > > > BabyAI setup contains about 40 tasks. Out of these 40 tasks, around 15 tasks are compositional i.e., requires to compose information from 2 or more sub-tasks, and rest of the tasks requires the agent to execute a particular behaviour (ex. going to the door, fetching a key etc).
> > > > > >
> > > > > > We study as to how often the agent retrieves information from other tasks. Ideally, for the compositional tasks, the hope would be the agent access information about the other tasks more often as compared to the tasks which require only a particular behaviour. So, during test time, we study the percentage of times agent access information from the same task as compared to accessing information from the different tasks. For compositional tasks, the agent access information from other tasks about  54% of times, while the percentage for single tasks is about 21\%. Individual numbers for compositional tasks are ((54\%, 43\%, 61\%, 75\%, 57\%, 32\%, 48\%, 66\%, 42\%, 51\%, 47\%, 65\%, 69\%, 51\%, 45\%). That said, we were not able to find any pattern as to find why the agent is retrieving more information from other tasks for some particular tasks as well as any structure about the states from which information is being retrieved.

---

> > > > > > > ### Author Response · Authors · 2021-11-23
> > > > > > > **Updated the paper**
> > > > > > >
> > > > > > > **I think an updated manuscript with these new elements would convince me to raise more.**
> > > > > > >
> > > > > > > Dear. Reviewer,
> > > > > > >
> > > > > > > Thanks again for giving us a change to improve our manuscript.
> > > > > > >
> > > > > > > These are the changes we have made to the manuscript:
> > > > > > >
> > > > > > > - Added the description of the episodic control baseline.
> > > > > > > - Added the results without the use of information bottleneck for multi-task BabyAI.
> > > > > > > - Added the discussion about the use of BERT style losses.
> > > > > > > - Added the discussion about positive results in Frostbite.
> > > > > > > - Added the discussion about the similar computation baseline (i.e., the baseline which does the same amount of computation but just not retrieve information).
> > > > > > > - Updated the use of LSTM and GRU.
> > > > > > > - Discussed details about the ablations and the role of top-k parameter.
> > > > > > >
> > > > > > > Thanks for your help and time. :)

---

> > > > > > > > ### Author Response · Authors · 2021-11-29
> > > > > > > > **Any further feedback ?**
> > > > > > > >
> > > > > > > > Dear. Reviewer,
> > > > > > > >
> > > > > > > > **With this in mind, I'm going to raise my score, but would need a bit more; I think an updated manuscript with these new elements would convince me to raise more.**
> > > > > > > >
> > > > > > > > We feel that your feedback has helped us in improving the paper. We would like feedback on the changes we have done since the comments from the reviewer, and would like to know if we can provide any other clarification to satisfy reviewer's concerns.
> > > > > > > >
> > > > > > > > Thanks a lot for your time, and help throughout the review process. We appreciate it.

---

> > > > > > > > > ### Comment · Reviewer_LSKp · 2021-11-30
> > > > > > > > > **Reply**
> > > > > > > > >
> > > > > > > > > Thanks for making those changes, which are a large part of why I chose to raise to a weak accept. However for this to be a stronger accept, the remaining points about the empirical evidence in my feedback were still not addressed in the updated manuscript. I was not convinced that the offline performance containing 30 `Gridroboman` tasks could not also be achieved using R2D2 (just a longer run time), and also did not see training curves for BabyAI. Furthermore things like significance testing were still not performed.

---

> > > > > > > > > > ### Author Response · Authors · 2021-11-30
> > > > > > > > > > **training curves for BabyAI**
> > > > > > > > > >
> > > > > > > > > > "also did not see training curves for BabyAI"
> > > > > > > > > >
> > > > > > > > > > As mentioned here (https://openreview.net/forum?id=0q0REJNgtg&noteId=NguMQmc6wc), BabyAI takes more than a week (to be precise ~ 10 days) to train and we don't have the saved logs as of now, and hence we were not able to update the paper.
> > > > > > > > > >
> > > > > > > > > > Thanks for your help and feedback. We appreciate it.

---

### Official Review · Reviewer_yAsc · 2021-10-31

**Correctness:** 2
**Technical Novelty And Significance:** 2
**Empirical Novelty And Significance:** 2
**Recommendation:** 5
**Confidence:** 3

**Main Review:**

Strengths:
The high-level idea of retrieving relevant information from past/expert experiences to help make decisions in the current states is very natural and well-motivated, though not very novel considering the existing works in experience replay and episodic memory.
The paper is well-organized and relatively easy to follow (there are several clarification issues, see the weakness part).
This work is a good trial to adapt the success of retrieval-based methods from language models to RL problems.

Weakness:

This work is closely related to episodic control, which also retrieves information from past experiences for value estimation. Could you add the prior work in episode control as a baseline? This will be helpful to confirm that introducing the complicated retrieval process is necessary rather than just keeping the simple episodic memory.

Could we use R2A in continuous control tasks? Based on my understanding the proposed method can be adapted to RL problems with continuous action. It will be great if there are additional experiments to verify this. Because most works in episode control require discrete action space, this could be one advantage of R2A if R2A can be used in continuous control tasks.

The retrieval process looks complicated (see the dense notations in section 2.2, 2.3), I'm generally concerned about why should we use such a complicated design? what's the importance of each component in the retrieval process? is it possible to simplify the design? The design choice might need additional explanations about the intuition or ablative studies about the effect of each component.
1. what's the intuition of using independent slots as the internal state of the retrieval process? why not just one slot? how is the information retrieved by each slot different from each other? Is the number of slots a sensitive hyper-parameter? How does not the number of slots influence the final performance?
2. What does the internal state m_t represent? It is clear that s_t represents the hidden state information in the MDP with partial observation. Then what's the intuition behind m_t? How is it different from s_t?
3. Information bottleneck is used to regularize the information retrieval. Is the final performance sensitive to the weight of this regularization loss?
4. Is the retrieval network optimized with temporal error in optimizing Q function? It is not crystal clear in the paper what is the overall loss term for the retrieval process.

The application of R2A on offline RL settings is interesting. Why not consider the standard offline RL benchmark (e.g. D4RL) and compare R2A with recent offline RL approaches (e.g. CQL, REM)? The experiment on commonly used benchmarks will be more convincing.

As pointed out in the first paragraph in section 3.2, one big challenge in offline RL is the distributional shift. It is unclear to me why the retrieval process can be helpful "particularly for states and actions that are rare in the offline dataset"? For rarely visited state-action pairs, the Q value estimation is hard, does the retrieval process make the Q value estimate more accurate on these state-action pairs?  It will be great to show the error in Q value estimation (i.e. difference between the ground truth Q value and estimated Q value) on simple discrete MDPs (where we can access the ground truth Q value). If the error becomes smaller after using R2A, such a claim will be further verified.

Looking forward to seeing stronger evidence to better support the advantages of the proposed method.




**Summary Of The Paper:**

This work proposes a retrieval neural network to help value-based RL algorithms (i.e. DQN and R2D2) retrieve information relevant to the current state from a dataset. The dataset may consist of the agent's own past experiences, expert demonstrations, or experiences from separate behavior policies (i.e. consider the offline RL setting). The retrieval network work is a recurrent model. The internal state in the retrieval network is partitioned into slots. Each slot independently retrieves information from the trajectory dataset. The retrieved information is used to update the internal state of the retrieval network and used as an additional input of the value function.

Experiment on Atari games verifies that the retrieval network improves the performance of R2D2. Also, this paper presents results on grid environments to verify that the retrieval network can be beneficial in offline RL settings.

**Summary Of The Review:**

The proposed method involves complicated design choices which are not well justified. Also, some important baselines are missing in the evaluation in the experiment part.

---

> ### Author Response · Authors · 2021-11-16
> **Results of R2A for continuous control tasks and more ablations (1/2)**
>
> We thank the reviewer for their time in reviewing our work and providing very useful feedback. We believe the reviewer's feedback is very constructive and has helped us in improving our work. We have ran additional experiments on continuous control benchmarks and performed ablations (we note that each of these experiments take approximately a week) to address the reviewer's main concerns.
>
> **This work is closely related to episodic control, which also retrieves information from past experiences for value estimation. Could you add the prior work in episode control as a baseline? This will be helpful to confirm that introducing the complicated retrieval process is necessary rather than just keeping the simple episodic memory.**
>
> The reviewer is indeed correct that the work is relevant to the episodic control literature (as also discussed in related work, section 4)
>
> The idea of allowing deep RL agents to adapt based on past experiences using a non-parametric memory is not new (Blundell et al., 2016; Pritzel et al., 2017; Hansen et al., 2018; Eysenbach et al., 2019; van Hasselt et al., 2019).
>
> There are two important differences between R2A and these methods:
>
> (1) In these methods, a local action-value function is constructed by using information about the nearest neighbors in the replay buffer, and then the agent makes a decision about which action to execute based on both the local value function as well as the global value function. However, in the proposed work, we employ a parameterized network (the retrieval process), which has access to the information in the replay buffer, and the agent process uses the retrieved information to shape the predictions of its value function in a fully differentiable way (using attention).
>
> (2) In these episodic control methods, there is only one process (the agent), which has direct access to the replay buffer. However, in R2A, the agent process only has indirect access to the replay buffer via the retrieval process.
>
> We have performed ablations where we compared to the scenario where we just keep the simple episodic memory (3.1.1 ABLATIONS AND ANALYSIS, ablation A-1). The result of ablations shows that for R2D2 accessing the dataset in a non-parametric way actually hurts the performance of the RL algorithm.
>
> **Could we use R2A in continuous control tasks?  Because most works in episode control require discrete action space, this could be one advantage of R2A if R2A can be used in continuous control tasks.**
>
> We thank the reviewer for suggesting this experiment.
>
> The design of R2A is agnostic of the RL algorithm. Specifically, we augment an RL agent with a retrieval process (parameterized as a neural network) that has direct access to a dataset of experiences. This dataset can come from the agent's past experiences, expert demonstrations, or any other relevant source. The retrieval process is trained to retrieve information from the dataset that may be useful in the current context, to help the agent achieve its goal faster and more efficiently.
>
> In order to address the reviewer's concern we performed additional experiments on the CausalWorld dataset (https://arxiv.org/abs/2010.04296). We choose CausalWorld because it is a benchmark for causal structure and transfer learning in a robotic manipulation environment. The environment is a simulation of an open-source robotic platform, hence offering the possibility of sim-to-real transfer. Tasks consist of constructing 3D shapes from a given set of blocks - inspired by how children learn to build complex structures. The key strength of CausalWorld is that it provides a combinatorial family of such tasks with common causal structure and underlying factors (including, e.g., robot and object masses, colors, sizes). The user (or the agent) may intervene on all causal variables, which allows for fine-grained control over how similar different tasks (or task distributions) are. We use the following four tasks: (a) Pushing (b) Picking ( c ) Pick and Place (d) Stacking2.  We follow the multi-task setting as mentioned in section 3.3.. Multi-task retrieval data can be either harmful if the retrieved information misguides the agent or beneficial if information from the other tasks is relevant to the current task. Since there’s some shared structure in the above selection of tasks, we hypothesize that R2A will be able to ignore distracting information and retrieve relevant information from other tasks.
> We followed the same experimental protocol as in CausalWorld, and used SAC as an RL algorithm. We train all the baselines and the proposed method for 100M environment steps in a multi-task fashion. We report the fractional success rate (i.e., as to how many times the agent is able to solve the task at hand).
>
> Multi-task SAC achieves a success rate of 73.5 +/- 3.6 \%, RA-RSAC (single-task retrieval buffer) achieve a success rate of 82.5 +/- 5%, RA-RSAC (multi-task retrieval buffer) achieves a success rate of 86.4 +/- 4.3\% (5 seeds).

---

> > ### Author Response · Authors · 2021-11-16
> > **Multiple slots, Training Procedure and standard offline RL benchmarks (2/2)**
> >
> > **What's the intuition of using independent slots as the internal state of the retrieval process? why not just one slot? how is the information retrieved by each slot different from each other? Is the number of slots a sensitive hyper-parameter? How does not the number of slots influence the final performance?**
> >
> > Deep learning has seen a movement away from representing examples with a monolithic hidden state towards a richly structured state. For example, Transformers segment by position, and object-centric architectures decompose images into entities. The induced structure, and separation of knowledge has improved generalization, model-size scaling, and long range dependencies. Hence, we follow the similar architecture where there are multiple positions or multiple slots. We performed an ablation where we reduced the number of slots to 1. We follow the same setup as in 3.1.1 ABLATIONS AND ANALYSIS. We ablate RA-R2D2 on the 10 Atari games it performs best relative to R2D2. RA-R2D2. RA-R2D2 mean improvement over R2D2 is around 72\%. On reducing the number of slots to 1, the mean improvement over R2D2 is around 27\%. We also note that having multiple slots allows the retrieval process to search for information in the dataset more efficiently (as different slots in parallel can search for relevant information).
> >
> > **What does the internal state m_t represent? It is clear that s_t represents the hidden state information in the MDP with partial observation. Then what's the intuition behind m_t? How is it different from s_t?**
> >
> > The retrieval process is parameterized as a neural network and has an internal state m_t. The retrieval process takes in the current abstract state of the agent process s_t and its own previous internal state m_{t−1} and uses these to retrieve relevant information from the dataset B, which it then summarizes in a vector u_{t}, and also updates its internal state m_{t}. In ablation section, we have shown the importance of having a separate internal state and also for the retrieving information (for more details: please refer to 3.1.1 ABLATIONS AND ANALYSIS)
> >
> > **Is the retrieval network optimized with temporal error in optimizing Q function? It is not crystal clear in the paper what is the overall loss term for the retrieval process**
> >
> > We thank the reviewer for pointing it out. We would improve the presentation. The two processes are trained in an end-to-end fashion. As discussed in A.6 LIMITATIONS AND FUTURE WORK (in appendix), It would be useful to investigate training of the retrieval process and the agent process using different objectives as compared to training them in an end-to-end fashion.
> >
> > **Why not consider the standard offline RL benchmark (e.g. D4RL) and compare R2A with recent offline RL approaches (e.g. CQL, REM)? The experiment on commonly used benchmarks will be more convincing**
> >
> > For our atari experiments, we build over the R2D2, which is a state of the art model-free baseline, used extensively in the RL algorithms (Agent57, NGU). For offline experiments, on varying the number of tasks (Figure 4),  we noticed that, the baseline DQN agent (blue) and the retrieval-augmented DQN agent (orange) perform identically for a single task setting; however, when the number of tasks increases (b, c), the retrieval-augmented agent learns much more effectively than the baseline DQN agent. Commonly used offlineRL benchmark (D4RL) mostly evaluates on single task setting, and hence we constructed our own multi-task offline RL benchmark (gridRoboman).

---

> ### Author Response · Authors · 2021-11-21
> **Further Clarifications ?**
>
> Dear Reviewer yAsc,
>
> Thanks for your time in reviewing our work. We have responded to your initial comments. We are looking forward to your feedback and are glad to answer your further questions. We would also like to thank as the experiment suggested by the reviewer helped us to convince another reviewer, and they increased their score.
>
> Thanks for your help, and time. We appreciate it.

---

> ### Author Response · Authors · 2021-11-26
> **Addressing  concerns ?**
>
> Dear Reviewer,
>
> Since the review period is coming to an end, we would really appreciate the opportunity to discuss further if our response has addressed your concerns. Many thanks!

---

> ### Comment · Reviewer_yAsc · 2021-11-29
> **Feedback after rebuttal**
>
> Thank the authors for the detailed feedback and additional experiments. It is great to see that the proposed method is beneficial for continuous control tasks. The rebuttal and updated paper partially address my concern about the architecture design of the retrieval process. However, an intuitive explanation about the effect of each component is still lacking in the retrieval process.  The evaluation of the proposed method in an offline setting can be improved if compared with SOTA offline RL approaches. Because the offline dataset is not a commonly used benchmark, it is unclear whether the advantage depends on the selected offline dataset or not. It is unclear whether the proposed method will keep an advantage if the offline dataset changes. Overall, I slightly adjust my rating.

---

> > ### Author Response · Authors · 2021-11-29
> > **Thanks for engaging.**
> >
> > We thank the reviewer for engaging with us and providing feedback, and thanks for increasing score.
> >
> > We would have liked more time, but we feel confident we can address rest of the reviewer's concern.
> >
> > **concern about the architecture design of the retrieval process**
> >
> > The idea behind the retrieval process is to search the retrieval dataset for information which may be relevant to the agent process in it's current state. The idea behind using multiple slots in the retrieval process is that different slots can *independently and in parallel* search for relevant information in the retrieval dataset (as compared to using a single slot). In the beginning of the episode different slots are randomly initialized, and compete with each to access different parts of the retrieval dataset.
> >
> > By different Ablations, we have shown the utility of following design choices:
> > - Importance of number of slots > 1
> > - Importance of retrieving information (same capacity and same memory baseline)
> > - Importance of the length of the contextual information in the retrieval dataset.
> >
> > **the offline dataset is not a commonly used benchmark**
> >
> > We agree. We note that none of the commonly used datasets have *multi-task* option, nor none of the baselines suggested by the reviewer have been tested in multi-task setting. The results in the paper shows that the proposed method performs better when we increase the number of tasks. That said, we did evaluate our method on scale on entire set of atari games, on BabyAI task, on synthetic offlineRL benchmark, on multi-task SAC, and show that the proposed method performs better across all these benchmarks.

---

### Official Review · Reviewer_cZ2E · 2021-11-02

**Correctness:** 3
**Technical Novelty And Significance:** 2
**Empirical Novelty And Significance:** 2
**Recommendation:** 5
**Confidence:** 4

**Main Review:**

The paper proposed a simple approach of end-2-end2 differentiable state-augmentation through retrieval.

Strenght:
* The approach is well written and the paper is pleasant to read.
* The modular choices composing the proposed method are reasonably clear and motivated.
* The experimental setting is clearly explained and reproducible.
* The experiment's motivations are clearly justified.

Weakness:
* The contribution is marginal, state augmentation is a reasonably classic topic and the end-2-end retrieval process could have been replaced by an attention model over the dataset of trajectories. We would have hoped to see this baseline studied deeply to justify the approach. In the introduction, the authors claim that a retrieval process allows simplifying the memorization duty of the model. We would have hoped to see experiments defending this claim.
* We would have hoped for a more exhaustive justification of why certain Atari games seem to improve significantly compared to others. it doesn't seem to have a general pattern of the game occurring where the approach is particularly beneficial.
* For offline RL, it would have been interesting to compare to SoA approaches, using for example the D4RL dataset which is dedicated to this task.

**Summary Of The Paper:**

The paper proposed a state-augmentation strategy based on a fully differentiable model for online and offline reinforcement learning.
The authors evaluate the proposal on two algorithms, DQN and R2D2.
The augmentation through retrieval approach is decomposed in several steps, namely trajectory, and state retrieval based on the currently perceived state and an internal state, the retriever has been defined with a recurrent neural network type of model.
The retrieved information is then aggregated using a so-called summarizing model which is also end-2-end differentiable.
The authors claim and show detailed improvement with respect to the state of the art on Atari and multi-task offline RL.
In the case of multi-task, the authors show that incorporating multiple tasks into the retrievable dataset is beneficial.

**Summary Of The Review:**

The paper proposed a retrieval-based state augmentation approach for online and offline reinforcement learning.
The retriever is stateful and end-2-end differentiable.
The approach is evaluated in online RL using Atari benchmark and offline RL.
The experiments in offline RL missed the main baselines and could have been evaluated on a state-of-the-art benchmark like the D4RL benchmark.
The retrieval approach also could have been evaluated against a fully attention-based model over the dataset, like Performer or Reformer Transformer architecture for example.
In conclusion, while the paper is reasonably well written, the contribution seems somehow limited and more experiments are needed to assess the full benefit of the proposed approach.

---

> ### Author Response · Authors · 2021-11-16
> **attention model over the dataset of trajectories + Continuous control RL tasks + offlineRL benchmarks**
>
> We thank the reviewer for their time in reviewing our work. We are enthused that the reviewer finds our work "very well written and the pleasant to read".
>
> **end-2-end retrieval process replaced by an attention model over the dataset of trajectories**
>
> The reviewer is right. One possible way to make use of the information in the dataset is direct attention over the dataset of trajectories as compared to indirect attention (amortized by a retrieval process). We note that we indeed performed this ablation where we compared to the scenario when we do a direct attention over dataset of experiences (3.1.1 ABLATIONS AND ANALYSIS, ablation A-1). The result of ablations shows that for R2D2 accessing the dataset in a non-parametric way actually hurts the performance of the RL algorithm. One possible conjecture is: When the agent has a direct access to the dataset, the agent has to search for the relevant information as well as learn an optimal policy, whereas when the agent has an indirect access to the dataset via retrieval process, the process of retrieving information is amortized by the retrieval process. This could also be a reason as to why the performance of the R2A improves on increasing the number of tasks (as for offline RL gridroboman experiments).
>
> Another intuition would be something like this: Imagine two processes agent and helper, where we assume the agent is following an unknown reward-maximizing policy, and you want to learn a helper process to maximize the agent's reward. for example, if you think about like a website, the agent will want to get to certain pages or get into certain states and if you imagine personalization as a "helper", it wants the user to accomplish those goals but faster and more easily even though user policy/reward are unknown like a user may search for a movie on netflix (which it must find rewarding), and one can see the recommender as a helper which lets the user get the reward faster and more reliably.
>
> **New experiments on continuous control RL tasks CausalWorld**
>
> As Reviewer yAsc, suggested we have ran more experiments for continuous control RL tasks.  We performed additional experiments on the CausalWorld dataset (https://arxiv.org/abs/2010.04296). We choose CausalWorld because it is a benchmark for causal structure and transfer learning in a robotic manipulation environment. The environment is a simulation of an open-source robotic platform, hence offering the possibility of sim-to-real transfer. Tasks consist of constructing 3D shapes from a given set of blocks - inspired by how children learn to build complex structures. The key strength of CausalWorld is that it provides a combinatorial family of such tasks with common causal structure and underlying factors (including, e.g., robot and object masses, colors, sizes). The user (or the agent) may intervene on all causal variables, which allows for fine-grained control over how similar different tasks (or task distributions) are. We use the following four tasks: (a) Pushing (b) Picking ( c ) Pick and Place (d) Stacking2. We follow the multi-task setting as mentioned in section 3.3.. Multi-task retrieval data can be either harmful if the retrieved information misguides the agent or beneficial if information from the other tasks is relevant to the current task. Since there’s some shared structure in the above selection of tasks, we hypothesize that R2A will be able to ignore distracting information and retrieve relevant information from other tasks. We followed the same experimental protocol as in CausalWorld, and used SAC as an RL algorithm. We train all the baselines and the proposed method for 100M environment steps in a multi-task fashion. We report the fractional success rate (i.e., as to how many times the agent is able to solve the task at hand).
>
> Multi-task SAC achieves a success rate of 73.5 +/- 3.6 %, RA-RSAC (single-task retrieval buffer) achieve a success rate of 82.5 +/- 5%, RA-RSAC (multi-task retrieval buffer) achieves a success rate of 86.4 +/- 4.3% (5 seeds).
>
> **could have used D4RL.**
>
> For atari experiments, we build over the R2D2, which is a state of the art model-free baseline, used extensively in the RL algorithms (Agent57, NGU). For offline experiments, on varying the number of tasks (Figure 4), we noticed that, the baseline DQN agent (blue) and the retrieval-augmented DQN agent (orange) perform identically for a single task setting; however, when the number of tasks increases (b, c), the retrieval-augmented agent learns much more effectively than the baseline DQN agent. Commonly used offlineRL benchmark (D4RL) mostly evaluates on single task setting, and hence we choose to construct our own multi-task offline RL benchmark (gridRoboman).
>
> **Performer or Reformer architecture**
>
> On BabyAI task, RA-RDQN (multi-task retrieval buffer) achieves 55% ± 5%, 74% ± 3% with 50K and 200K trajectories. Reformer architecture achieves 41% ± 3% and 55% ± 6% (much worse than the proposed method).

---

> > ### Author Response · Authors · 2021-11-21
> > **Ablation results**
> >
> > **working better on atari games**
> >
> > As mentioned in “Building Machines That Learn and Think Like People”, in Frostbite, players control an agent (Frostbite Bailey) tasked with constructing an igloo within a time limit. The igloo is built piece-by-piece as the agent jumps on ice floes in water The challenge is that the ice floes are in constant motion (moving either left or right), and ice floes only contribute to the construction of the igloo if they are visited in an active state (white rather than blue). The agent may also earn extra points by gathering fish while avoiding a number of fatal hazards (falling in the water, snow geese, polar bears, etc.). Success in this game requires a temporally extended plan to ensure the agent can accomplish a sub-goal (such as reaching an icefloe) and then safely proceed to the next sub-goal. Ultimately, once all of the pieces of the igloo are in place, the agent must proceed to the igloo and thus complete the level before time expires. One hypothesis for the better performance could be that the retrieval process is able to efficient utilize the information from the states which are very far apart from the current state (i.e., temporally extended credit assignment). This hypothesis is further validated when we decrease the length of the traces in the dataset. In section 3.1.1 ABLATIONS AND ANALYSIS, in ablation (Shorter retrieved trajectories (A-3)), we decrease the length of the trajectories that are retrieved and summarized during retrieval preprocessing, thus reducing the amount of past and future information the retrieval process can retrieve. By default, the trajectories in the retrieval dataset are of length 80. To perform this ablation, we decrease the length of the effective context to only include information from 5 timesteps.
> >
> >
> >
> > **Ablation studying how the retrieval process is working**
> >
> > We tried to conduct relevant ablations to understand the different components of the proposed architecture:
> >
> > Specifically, we conducted these ablations:
> > - Importance of a separate retrieval process (A-1). In R2A, the retrieval process and the agent process are parameterized separately,
> > i.e., they have their own internal states. Here we examine what happens when the agent’s state is used to query the retrieval batch
> > instead of using the retrieval state (normal episodic control baseline).
> >
> > - Importance of retrieving information (A-2). We examine what happens when the retrieval process does not have access to the retrieval
> > dataset and hence no information is retrieved (same computation baseline)
> >
> > - Shorter retrieved trajectories (A-3). We decrease the length of the trajectories that are retrieved and summarized during retrieval preprocessing.
> >
> > - Importance of auxiliary losses to summarize retrieval batch (A-4, A-5). Here we study the use of self-supervised BERT style masking losses in addition to using action, reward and value prediction.
> >
> > - Role of Topk (in appendix): In appendix, A.3.3 ADDITIONAL ATARI ABLATIONS, where we perform ablations on RA-R2D2 using the 5 Atari games on which RA-R2D2 performs worst relative to baseline R2D2. In our experiments, the retrieval process selects the top k_traj = 10 most relevant trajectories (step 2, section 2.3) and then selects the top k_states = 10 most relevant states of these trajectories (step 3, section 2.3) from which to retrieve relevant information. To better understand the role of these hyperparameters, we independently vary these two hyper-parameters (top k_traj and top k_states) over the values {5, 10, 20}. Figure 10 shows that after independently varying these two hyper-parameters the performance of the R2A can be improved as compared to the R2D2 baseline.

---

> ### Author Response · Authors · 2021-11-22
> **Further Clarifications ?**
>
> Dear Reviewer cZ2E,
>
> Thanks for your time in reviewing our work. We have responded to your initial comments. We are looking forward to your feedback and are glad to answer your further questions. We would also appreciate feedback on the experiments conducted as mentioned in your review (comparison to Reformer), and more results on continuous control benchmark (CausalWorld).
>
> Thanks for your help, and time. We appreciate it.

---

> ### Author Response · Authors · 2021-11-28
> **Feedback on extra experiments ?**
>
> Dear. Reviewer,
>
> We did extra experiments as asked by the reviewer. We would appreciate feedback on our rebuttal. We would also be happy to clarify any other concerns which could help the reviewer improve the understanding of the paper.

---

### Official Review · Reviewer_YWJ5 · 2021-11-07

**Correctness:** 3
**Technical Novelty And Significance:** 3
**Empirical Novelty And Significance:** 3
**Recommendation:** 6
**Confidence:** 4

**Main Review:**

Overall the approach is well-motivated and is clear in its presentation. The results are strong to demonstrate the utility of the proposed approach.

The following are some of my questions and would be great if the authors could clarify some of them:

1. The retrieval process relies on auxiliary losses based on reward, value predictions and on BERT-style losses. Could the authors present the exact loss functions for training these modules in the main text? I was not able to find any details related to how these modules were trained and it is an important detail relevant to the approach.

2. Because the main contribution of the paper is an empirical improvement across domains, would the authors release the code or a very-detailed pseudocode (similar to the one presented in Schrittwieser et al. 2020), which seems necessary for a reimplementation?

Schrittwieser J, Antonoglou I, Hubert T, Simonyan K, Sifre L, Schmitt S, Guez A, Lockhart E, Hassabis D, Graepel T, Lillicrap T. Mastering atari, go, chess and shogi by planning with a learned model. Nature. 2020 Dec;588(7839):604-9.

3. An ablation studying how the retrieval process is working is necessary. Intuitively it makes sense that the retrieval process is considering prior experiences to make an accurate estimate of the Q-values. For example, in the umbrella domain (Harutyunyan et al. 2019), an agent picking up an umbrella much earlier in an episode is useful information to have to make an accurate estimate of Q-values for later parts of the episode. Thus, I would find it interesting to see if the retrieval process makes similar computations.

Harutyunyan, A., Dabney, W., Mesnard, T., Gheshlaghi Azar, M., Piot, B., Heess, N., ... & Munos, R. (2019). Hindsight credit assignment. Advances in neural information processing systems, 32, 12488-12497.

4. Could the authors, in addition Fig 2, present the Atari performance as a learning curve where the median-normalized performance is reported as a function of training steps? This would tell if the agent overall produces a faster learning performance and also tells the final median performance across all atari games.


**Summary Of The Paper:**

The paper presents an approach that augments an RL agent with a parameterized retrieval process directly accessing prior experiences. The retrieval process is trained to retrieve from the dataset based on the current context of the agent and thus make a better prediction of the Q-values. The approach is called Retrieval Augmented RL (RARL) and is demonstrated on Atari domains.


**Summary Of The Review:**

Overall I think the paper makes an interesting contribution to RL where the agent is able to access prior data in making accurate estimations of Q-values and through that produce a better policy for maximizing rewards from a given task.

Some concerns related to the paper are as follows: Providing detailed pseudocode or code for replicating the results, ablation to understand the retrieval process, and detailed presentation of the proposed approach.

---

> ### Author Response · Authors · 2021-11-16
> **Ablations + Auxiliary losses + CausalWorld experiments**
>
> We thank the reviewer for taking time and reviewing our work. We appreciate the thorough feedback. We are enthused that the reviewer finds the paper to be well written.
>
> **Relies on auxiliary losses based on reward predictions and on BERT-style losses.**
>
> For reward, value predictions, we take the current abstract representation, and use it to predict rewards, values and actions (in a supervised way). These losses are computed the same way as in MuZero (except we don't unroll the latent state). We would add a description of it in the paper.
>
> For BERT style auxiliary losses, we tokenize the representation at different time-steps, and then mask 15\% of the time-steps, and using the un-masked time-steps we predict the representation at the masked positions.
>
> **a very-detailed pseudocode**
>
> We have written a detailed pseudo code (as in the algorithm box). We ask the reviewer to refer to:  Algorithm 1 One timestep of a retrieval-augmented agent (R2A). Please let us know if the reviewer wants us to add more details. Thanks for your help and time. We appreciate it.
>
> **Ablation studying how the retrieval process is working**
>
> We tried to conduct relevant ablations to understand the different components of the proposed architecture:
>
> Specifically, we conducted these ablations:
> - Importance of a separate retrieval process (A-1). In R2A, the retrieval process and the agent process are parameterized separately,
> i.e., they have their own internal states. Here we examine what happens when the agent’s state is used to query the retrieval batch
> instead of using the retrieval state (normal episodic control baseline).
>
> - Importance of retrieving information (A-2). We examine what happens when the retrieval process does not have access to the retrieval
> dataset and hence no information is retrieved (same computation baseline)
>
> - Shorter retrieved trajectories (A-3). We decrease the length of the trajectories that are retrieved and summarized during retrieval preprocessing.
>
> - Importance of auxiliary losses to summarize retrieval batch (A-4, A-5). Here we study the use of self-supervised BERT style masking losses in addition to using action, reward and value prediction.
>
> - Role of Topk (in appendix): In appendix, A.3.3 ADDITIONAL ATARI ABLATIONS, where we perform ablations on RA-R2D2 using the 5 Atari games on which RA-R2D2 performs worst relative to baseline R2D2. In our experiments, the retrieval process selects the top k_traj = 10 most relevant trajectories (step 2, section 2.3) and then selects the top k_states = 10 most relevant states of these trajectories (step 3, section 2.3) from which to retrieve relevant information. To better understand the role of these hyperparameters, we independently vary these two hyper-parameters (top k_traj and top k_states) over the values {5, 10, 20}. Figure 10 shows that after independently varying these two hyper-parameters the performance of the R2A can be improved as compared to the R2D2 baseline.
>
>
> **Experiments on continuous control RL tasks CausalWorld**
>
> As Reviewer yAsc, suggested we have ran more experiments for continuous control RL tasks, on the CausalWorld dataset (https://arxiv.org/abs/2010.04296). We choose CausalWorld because it is a benchmark for causal structure and transfer learning in a robotic manipulation environment. The environment is a simulation of an open-source robotic platform, hence offering the possibility of sim-to-real transfer. Tasks consist of constructing 3D shapes from a given set of blocks - inspired by how children learn to build complex structures. The key strength of CausalWorld is that it provides a combinatorial family of such tasks with common causal structure and underlying factors (including, e.g., robot and object masses, colors, sizes). The user (or the agent) may intervene on all causal variables, which allows for fine-grained control over how similar different tasks (or task distributions) are. We use the following four tasks: (a) Pushing (b) Picking ( c ) Pick and Place (d) Stacking2. We follow the multi-task setting as mentioned in section 3.3.. Multi-task retrieval data can be either harmful if the retrieved information misguides the agent or beneficial if information from the other tasks is relevant to the current task. Since there’s some shared structure in the above selection of tasks, we hypothesize that R2A will be able to ignore distracting information and retrieve relevant information from other tasks. We followed the same experimental protocol as in CausalWorld, and used SAC as an RL algorithm. We train all the baselines and the proposed method for 100M environment steps in a multi-task fashion. We report the fractional success rate (i.e., as to how many times the agent is able to solve the task at hand).
>
> Multi-task SAC achieves a success rate of 73.5 +/- 3.6 %, RA-RSAC (multi-task retrieval buffer) achieves a success rate of 86.4 +/- 4.3% (5 seeds).

---

> > ### Comment · Reviewer_YWJ5 · 2021-11-29
> > **Thank you for the detailed rebuttal**
> >
> > I would like to thank the authors for a detailed rebuttal and follow-up experiments. I have read the other reviews and the responses to them. The rebuttal addresses most of my concerns.
> >
> > The ablations do a good job of demonstrating the role of each design choice used in the learning method. However, I still think visualizing the retrieval process in a small experiment would have been useful to see to clarify if the method is indeed retrieving from transitions as expected.
> >
> > With the current version of the paper, I am still happy with the presented results and would like to keep my score of weak acceptance unchanged.

---

> > > ### Author Response · Authors · 2021-11-29
> > > **method is indeed retrieving from transitions**
> > >
> > > Dear Reviewer,
> > >
> > > We thank the reviewer for taking time to reply and engage. We indeed did some experiments to see if the method is indeed retrieving information from transitions as expected.
> > >
> > > **a small experiment would have been useful to see to clarify if the method is indeed retrieving from transitions as expected.**
> > >
> > > As  Reviewer LSKp asked, we did some analysis to study the properties about the retrieved information in the multi-task setting in BabyAI.
> > > BabyAI setup contains about 40 tasks. Out of these 40 tasks, around 15 tasks are compositional i.e., requires to compose information from 2 or more sub-tasks, and rest of the tasks requires the agent to execute a particular behaviour (ex. going to the door, fetching a key etc).
> > >
> > > We study as to how often the agent retrieves information from other tasks. Ideally, for the compositional tasks, the hope would be the agent access information about the other tasks more often as compared to the tasks which require only a particular behaviour. So, during test time, we study the percentage of times agent access information from the same task as compared to accessing information from the different tasks. For compositional tasks, the agent access information from other tasks about 54% of times, while the percentage for single tasks is about 21%. Individual numbers for compositional tasks are (54%, 43%, 61%, 75%, 57%, 32%, 48%, 66%, 42%, 51%, 47%, 65%, 69%, 51%, 45%).
> > >
> > > We have also updated this in the paper.
> > >
> > > Would this help the reviewer to increase their score ?

---

### Decision · Program_Chairs · 2022-01-20

**Decision:**

Reject

**Comment:**

One of the four reviewers failed to engage in discussion, two acknowledged the author's response and paper revision without changing their scores, and one reviewer engaged in considerable discussion resulting in a score increase to a weak accept.  No reviewer gave the paper a strong endorsement.  I do appreciate the large effort that the authors put into revising their paper and addressing reviewers concerns.  However, major post-submission revision puts an inappropriate burden on reviewers.  In any case, there is not strong support for this paper even from the one heavily engaged reviewer.